# Tree Attention: Topology-Aware Decoding for Long-Context Attention on GPU Clusters

## Abstract

Self-attention is the core mathematical operation of modern transformer architectures and is also a significant computational bottleneck due to its quadratic complexity in the sequence length. In this work, we derive the scalar energy function whose gradient computes the self-attention block, thus elucidating the theoretical underpinnings of self-attention. Our formulation reveals that the reduction across the sequence axis can be efficiently computed in parallel through a tree reduction. Our algorithm, called `Tree Attention`, for parallelizing exact attention computation across multiple GPUs enables cross-device decoding to be performed *asymptotically* faster (up to 8× faster in our experiments) than state-of-the-art approaches such as `Ring Attention`, while also requiring significantly less communication volume and incurring 2× less peak memory. We demonstrate that `Tree Attention` speeds up decoding up to 4x on Llama 3.1-8B and can be applied to a variety of hardware and networking setups such as H100 DGX nodes, AMD MI300x nodes, and PCIe connected NVIDIA RTX 4090s. Our code is publicly available here: `https://anonymous.4open.science/r/tree_attention-7C32`

## 1 Introduction

The self-attention operation is the core computational building block of the transformer architecture Bahdanau et al. (2014); Vaswani et al. (2017), which has become an ubiquitous and highly effective workhorse architecture currently applied at scale to language Brown et al. (2020); Kaplan et al. (2020); Hoffmann et al. (2022); Team et al. (2023); Achiam et al. (2023); Pilault et al. (2023), vision Dosovitskiy et al. (2020), audio Betker (2023), and decision-making Chen et al. (2021); Reed et al. (2022). Nonetheless, the quadratic time complexity of self-attention means that significant resources are required to train and generate from transformer-based Large Language Models (LLMs), especially for models with large context lengths.

During inference, the attention block largely determines the computational and memory requirements, which become more demanding as the input sequence length increases. Although LLMs generate one token at a time, the entire sequence of past tokens must still be stored in memory and used to compute attention scores during generation. Since attention performs a similarity matching of every token representation with every other, it incurs quadratic computational complexity in terms of flops.

There have been recent advances in training LLMs to handle extremely long contexts (up to 1M tokens) Chen et al. (2023); kai (2023); Peng et al. (2023). Such models attain qualitatively new capabilities such as extremely large-scale in-context learning of entire small datasets held in the prompt Reid et al. (2024); Lee et al. (2024); Bertsch et al. (2024). They can also avoid putting multi-modal continuous data through a lossy tokenization scheme Reid et al. (2024); Team (2024) by directly operating at the byte level Xue et al. (2022); Wu et al. (2024). The issue however is that performing inference on such long contexts is very expensive.

To speed up inference and alleviate memory requirements, recent works have attempted to alter the attention mechanism itself, either by linearizing it Katharopoulos et al. (2020), or approximating it by a kernel map Choromanski et al. (2020b); Peng et al. (2021); Arora et al. (2024), which reduces the complexity to linear at the cost of reduced expressiveness. Others have invented alternative sequence mixing architectures such as state-space models which are designed to be efficiently computable in

linear time and constant memory Gu & Dao (2023); Dao & Gu (2024); Katsch (2023); Sun et al. (2023); Glorioso et al. (2024).

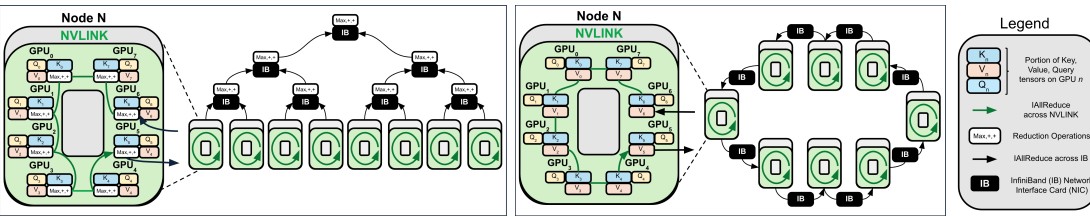

(a) Multi Node `Tree Attention` (Ours)  (b) Multi Node `Ring Attention`

Figure 1: Ring and `Tree Attention` Topologies. Due to the associative properties of the logsumexp and max operations of `Tree Attention` (Fig. 1(a)), is possible to structure the reduction across the sequence as a tree, requiring asymptotically fewer communication steps than `Ring Attention` (Fig. 1(b)) as well as less memory and communications volume.

Other approaches utilize efficient algorithms to reduce the computational burden of attention while keeping the core computation the same. These include memory-efficient attention Rabe & Staats (2021), `Flash Attention` Dao et al. (2022) and `Flash Decoding` fla (2024), which provide a set of IO-aware kernels to map the attention operation to the GPU hardware resources in an extremely efficient way, significantly reducing the memory overhead required. Further works Character AI (2024); Kang et al. (2024); Liu et al. (2024); Nawrot et al. (2024) explore compressing or otherwise reducing the KV cache required in generation. Finally, `Ring Attention` Liu et al. (2023) proposes a way to parallelize the attention computation across the sequence axis between GPUs, thus enabling significantly longer contexts than can be served on a single GPU. Since our proposed method is an exact calculation of attention[1], it is a plugin replacement for any multi-GPU sequence parallel mechanism such as the state of the art `Ring Attention` mechanisms. By leveraging the exact energy function for the self-attention block, we develop a method to speed up inference for long context use-cases when keys and values are sharded across multiple GPUs along the sequence axis.

Our proposed algorithm for computing attention via the gradient of the energy function is built on top of an efficient parallel computation and tree reduction communication strategy. In particular, this formulation lets us devise an asymptotically faster algorithm for performing decoding in which the number of communication steps scales logarithmically with the number of devices, instead of linearly in alternatives such as `Ring Attention` Liu et al. (2023). Our topology-aware approach illustrated in Fig. 1 significantly outperforms leading attention parallelization methods such as `Ring Attention` on multiple devices.

## 2  RELATED WORKS

The computational complexity of self-attention, introduced by Vaswani et al. (2017), poses challenges for long sequences due to its quadratic dependency on sequence length, $O(n^2 \cdot d)$. To address this, attention **approximation** mechanisms like Linformer (Wang et al., 2020) and Performer (Choromanski et al., 2020a) reduce complexity to linear $O(n)$ using low-rank projections and kernelized approximations on a *single device*. Sparse models such as Longformer (Beltagy et al., 2020) and BigBird (Zaheer et al., 2020) further optimize computations by restricting attention to local windows or sparsity patterns, significantly reducing resource demands while maintaining performance for specific tasks. Such methods however provide approximations to the attention mechanism while we seek to parallelize an **exact** attention computation across the sequence axis.

Theoretical work has also contributed to improving the efficiency of both exact and approximate methods. Kernel-based approaches, such as those by Tsai et al. (2019), suggest alternative formulations to self-attention that are computationally efficient. Surveys like Tay et al. (2020) highlight these advancements, emphasizing the synergy between parallelization strategies and sparsity or approximation techniques, ensuring self-attention remains scalable even in constrained computational environments. It must be noted as well that Duman Keles et al. (2022) established lower bounds

---

[1]It can be shown empirically that Ring Attention and Tree Attention are exact computations of Attention since both methods have exactly the same activations as the forward pass of Vanilla Attention.

on the computational complexity of self-attention, demonstrating that achieving sub-quadratic time complexity is unlikely unless the Strong Exponential Time Hypothesis (SETH) is false.

In addition to approximation methods, several approaches focus on parallelizing exact attention computations. `FlashAttention` (Dao et al., 2022), for instance, reorganizes the attention computation into smaller, memory-efficient blocks that leverage GPU memory hierarchies to enable faster and parallelized processing of exact attention. Other techniques use optimized matrix operations and tiling strategies to distribute attention computations across cores or threads efficiently (Shen et al., 2021). While these methods aim to maximize throughput while maintaining the precision of exact attention, they focus on speeding up single-device attention computation. Since we parallelize exact attention across multiple devices, `Ring Attention` (Liu et al., 2023) is the most comparable to our work. Finally, to the best of our knowledge, there are no other techniques that explore multi-device parallel decoding as we have.

## 3 SELF-ATTENTION

The self-attention operation can be represented as a set of dot product similarity searches between queries and keys. These similarity scores are then reduced along the sequence axis and softmaxed, so that for a given query, there is a probability distribution of the similarities of each given key. We then take the expectation of the value vectors against this distribution. We denote the queries assigned to a sequence of length $N$ as $\{q_a, a = 1, \cdots N\}$, where each query is a vector of size $d$ that stands for hidden dimension, $q_a \in \mathbb{R}^d$, and similarly the keys and values $\{(k_a, v_a), a = 1, \cdots N\}$. Attention can be written as

$$z^a = \sum_{i=1}^{N} \text{softmax}(q_a \cdot k_i^T) v_i .$$

Naively computing attention in this way requires materializing the $qk$ matrix with computational and memory cost quadratic in the sequence length. Memory-efficient attention Rabe & Staats (2021) is an iterative way to compute the softmax similarities without ever having to materialize the full attention matrix. It performs the following operations, one query (or a chunk of queries) at a time:

$$s_i^{(j)} = \exp(q_j \cdot k_i) \tag{1}$$

$$n_i^{(j)} = n_{i-1}^{(j)} + v_i s_i^{(j)} \tag{2}$$

$$d_i^{(j)} = d_{i-1}^{(j)} + s_i^{(j)} \tag{3}$$

Then, once the values $v$ and softmax denominator $d$ are computed, we divide to get the final softmaxed scores $z^{(j)} = \frac{n^{(j)}}{d^{(j)}}$ for every query index $j$. Computing attention in this iterative manner significantly reduces the required memory.

`Flash Attention` Dao et al. (2022) utilizes a similar approach to reduce the memory and computational cost of attention, but the algorithm is not adapted for multi-GPU computation. `Flash Attention` performs the iterative algorithm of Rabe & Staats (2021) in a blockwise manner, utilizing the block-parallel computational primitives available inside single GPU tensor cores. Additionally, it precisely sizes the blocks such that they can fit into the SRAM of the GPU for the entire attention computation, effectively performing kernel fusion and preventing many unnecessary IO operations.

## 4 SELF-ATTENTION AS THE GRADIENT OF AN ENERGY FUNCTION

Following the ubiquitous success of the transformer architecture, there has been significant effort to mathematically understand the nature and meaning of the attention operation and link it to energy models (Krotov & Hopfield, 2016; Krotov, 2021; Millidge et al., 2022; Hoover et al., 2024), such as Hopfield Networks (Ramsauer et al., 2020; D'Amico & Negri, 2024). Ramsauer et al. (2020) pioneered this field by performing a similar but distinct analysis to relate self-attention with the modern Hopfield networks, providing a novel and insightful interpretation of self-attention as performing hetero-associative memory lookups using a high-powered nonlinear similarity function. This work was later extended by Hoover et al. (2023), who derived a modified version of the transformer based off an energy function. However, while it has long been known that the softmax operation can be

derived as the gradient of the following scalar function:

$$\partial_{z_j} \log \sum_{a=1}^{n} \exp(z_a) = \frac{e^{z_j}}{\sum_{a=1}^{n} e^{z_a}} = \text{softmax}(z_j), \tag{4}$$

known as the log-sum-exp, an equivalent function for the self-attention block has not yet been derived. We develop in this paper a link between attention and energy functions by introducing an auxiliary *source* vector $\zeta$, which represents the "external contributions" to the system's energy (Hopfield, 1982). The *source* $\zeta$ is the parameter with respect to which we compute the gradient of the scalar energy function to obtain the self-attention operation. As we will see, we need the source in order to write down the generating function of the moments of the distribution since taking the gradient with respect to $\zeta$ yields the exact self-attention operation.

This insight allows us to make the following observation:

**Observation 1.** *Attention can be expressed at the gradient of an scalar energy function $F(\zeta)$ with respect to the source $\zeta$, such that:*

$$\sum_{a=1}^{N} \text{softmax}(q \cdot k_a) v_a = \left. \frac{\partial F}{\partial \zeta} \right|_{\zeta=0}, \tag{5}$$

*where the moment generating function (i.e. the energy function) $F(\zeta)$ is defined as:*

$$F(\zeta) = \log \sum_{a} \exp\left( q \cdot k_a^T + \zeta \cdot v_a^T \right). \tag{6}$$

The proof of Observation 1 can be found in Appendix C.1. Please note that this formulation also allows to make a Bayesian interpretation of Attention in Appendix C.2 and motivates our Tree Attention algorithm in the next Section 5.

## 5 TREE ATTENTION

In this section we show how the formulation of the attention operation as the gradient of an energy function suggests an efficient parallel strategy for computing it. The key insight is to leverage an efficient algorithm to compute the energy, and then differentiate it in order to obtain an efficient algorithm to compute attention.

### 5.1 EFFICIENT ENERGY FUNCTION COMPUTATION

Let us focus on the case of decoding with a KV cache in a causal language model where we have one query and $N$ keys and values. In this case, the energy function is:

$$F_{dec} = \log \sum_{a=1}^{N} \exp(q \cdot k_a^T + \zeta \cdot v_a^T) \equiv \text{logsumexp}_a(\{q \cdot k_a^T + \zeta \cdot v_a^T, a = 1, \cdots, N\}). \tag{7}$$

A crucial fact is that both $\text{logsumexp}_a$ and $\max_a$ are associative operations:

$$\text{logsumexp}_a(\{T_a, \text{logsumexp}_a(\{R_a, S_a\})\}) = \text{logsumexp}_a(\{\text{logsumexp}_a(\{T_a, R_a\}), S_a\}),$$

$$\max_a(\{\max_a(\{T_a, R_a\}), S_a\}) = \max_a(\{T_a, \max_a(\{R_a, S_a\})\}).$$

We can prove that this associative property allows these reductions to be performed efficiently in parallel with logarithmic time complexity, provided we have adequately many parallel workers:

**Theorem 1.** *The time complexity of a reduction operation involving an associative function, such as $\text{logsumexp}_a$ or $\max_a$, over an array of size N using p parallel processors is $O\left(\frac{N}{p} + \log p\right)$. When the number of processors p is equal to N, the time complexity is reduced to $O(\log N)$.*

The proof of Theorem 1 is in Appendix E.

Putting this result together, and for $\hat{a}, \hat{b} \in \{1, \cdots, t\}$ intra-chunk indices, we get the following highly parallel Algorithm 1:

---

**Algorithm 1** Single Query Energy Forward (calculating logsumexp)

---

1: Divide $\mathbf{k}, \mathbf{v} \in \mathbb{R}^{N \times d_h}$ into $p$ chunks $\{\mathbf{k}_{\hat{a}}, \mathbf{v}_{\hat{a}}, \hat{a} \in \{1, \cdots, N/p\}\}$ of size $t = N/p$
2: Scatter a copy of $\mathbf{q}, \zeta$, and each $\mathbf{k}_{\hat{a}}, \mathbf{v}_{\hat{a}}$ to each of the $p$ processors.
3: In parallel compute $r_{\hat{a}} = \mathbf{q} \cdot \mathbf{k}_{\hat{a}}^T + \zeta \cdot \mathbf{v}_{\hat{a}}^T$
4: Compute $m = \text{Reduce}(\max, r_{\hat{a}})$ by doing a tree reduction.
5: Scatter $m$ to every device and update $r_{\hat{a}} \rightarrow r_{\hat{a}} - m$.
6: Compute $lse = \text{Reduce}(\text{logsumexp}, r_{\hat{a}})$ by doing a tree reduction.
7: Save $lse, m$ for gradient w.r.t $\zeta$.
8: Return $lse$

---

## 5.2 Efficient parallel decoding

One of the core insights of automatic differentiation is that the gradient of a function $\nabla_x f(x)$ can be computed with the same time complexity as computing $f(x)$ Vieira (2016). The caveat however is that if the function has a deep computational graph, then the memory footprint of computing the gradient grows with that depth as backpropagation requires storing the values of the intermediate tensors. In our case, the computational graph involved in computing the energy is shallow and therefore the memory overhead is negligible. This means that if we can compute the energy efficiently, we obtain an efficient algorithm for computing its gradient (i.e. the self-attention operation) automatically.

In our case, we want to compute the gradient of the energy function with respect to $\zeta_A$ and then set it to zero. This can be done with automatic differentiation engines having set $\zeta$ to be a tensor of zeros from the very outset. We can however manually implement a gradient with respect to $\zeta$ pass of the above Algorithm 1 that does not materialize $\zeta$ in Algorithm 2 below. Note in particular that when we set $\zeta_A = 0$, $A \in \{1, \cdots, d_h\}$ then $lse$ involves only the logsumexp of the dot product between queries and keys.

---

**Algorithm 2** `Tree Decoding` (using atomic operation on each device)

---

1: Divide $\mathbf{k}, \mathbf{v} \in \mathbb{R}^{N \times d_h}$ into $p$ chunks $\{\mathbf{k}_{\hat{a}}, \mathbf{v}_{\hat{a}}, \hat{a} \in \{1, \cdots, N/p\}\}$ of size $t = N/p$
2: Calculate $m$ and $lse$ using Algorithm 1.
3: Scatter a copy of $\mathbf{q}, m$ and $lse$, and each $\mathbf{k}_{\hat{a}}, \mathbf{v}_{\hat{a}}$ to each of the $p$ processors.
4: In parallel compute $r_{\hat{a}} = \mathbf{q} \cdot \mathbf{k}_{\hat{a}}^T - m$
5: Compute $R_{\hat{a}} = \frac{\exp(r_{\hat{a}})}{\exp(lse)} \cdot v_{\hat{a}} = \exp(r_{\hat{a}} - lse) \cdot v_{\hat{a}}$
6: Compute $z = \text{Reduce}(\text{sum}, R_{\hat{a}})$
7: Return $z$

---

Notice here that by storing $lse, m$ for the backward pass, the only remaining reduction operation that needs to be performed is the one in line 5 of the above algorithm. This single reduction takes $O(N/p)$ time to compute the local sums on each device and $\log p$ time to communicate and combine partial results, and therefore we get the same asymptotic complexity as the logsumexp calculation.

In practice, we implement the forward and gradient w.r.t. $\zeta$ in a single function which returns both the value and the gradient of the energy function. We can therefore put together Algorithms 1 and 2 into the following efficient parallel decoding Algorithm 3:

---

**Algorithm 3** `Tree Decoding` (using `Flash Attention 2` on each device)

---

1: Divide $\mathbf{k}, \mathbf{v} \in \mathbb{R}^{N \times d_h}$ among $p$ GPUs, each with a chunk $\{\mathbf{k}_{\hat{a}}, \mathbf{v}_{\hat{a}}, \hat{a} \in \{1, \cdots, N/p\}\}$ of size $t = N/p$ and scatter $\mathbf{q}$ to each GPU.

2: Use `Flash Attention 2` to compute $\text{o} = \frac{\sum_{\hat{a}} \exp(\mathbf{q} \cdot \mathbf{k}_{\hat{a}}^T) \mathbf{v}_{\hat{a}}}{\sum_{\hat{b}} \exp(\mathbf{q} \cdot \mathbf{k}_{\hat{b}}^T)}$ and $\text{lse} = \log \sum_{\hat{b}} \exp(\mathbf{q} \cdot \mathbf{k}_{\hat{b}}^T)$.

3: Recompute the global max $m = \texttt{Allreduce}(\max, \text{lse})$.

4: Get local numerator and denominator by computing: $\text{n} = \text{o} * \exp(\text{lse} - m), \text{d} = \exp(\text{lse} - m)$.

5: Compute global numerator and denominator with: $\text{n}_g = \texttt{Allreduce}(\text{sum}, \text{n}), \text{d}_g = \texttt{Allreduce}(\text{sum}, \text{d})$.

6: Return result $z = \frac{\text{n}_g}{\text{d}_g}$.

---

This algorithm requires three `Allreduce` operations in total, meaning that the required time complexity is $O(3(N/p + \log p))$.

### 5.3 EFFICIENT COLLECTIVE OPERATIONS USING TOPOLOGY-AWARENESS

**Communication overheads** While the theoretical analysis above indicates that we should see speedups when using tree-based reductions, this is not necessarily guaranteed in practice due to various potential overheads. In particular, our argument for the time complexity of our proposed `Tree Decoding` algorithm assumes that communication of partial results is instantaneous, which in practice is never the case. In fact, as we scale the sequence length, or the number of GPUs especially to the multi-node setting, the time taken for communication is the dominant contribution to the total execution time. However, importantly, beyond its asymptotic benefits, `Tree Attention` benefits from taking advantage of the two-level topology which is standard in modern GPU clusters.

We benchmark our algorithm against a previously proposed sequence parallel attention algorithm called `Ring Attention`. Like our algorithm, `Ring Attention` assumes that the sequence is sharded across GPUs and performs the attention computation without gathering all of the sequence on to a single device. Instead, it communicates shards of the keys and values in a point-to-point manner between neighboring GPUs that are logically arranged in a ring topology. This communication is overlapped with the computation of the local shard of the output. In contrast with this strategy, our algorithm scatters the query and communicates the partial result across all GPUs when performing the `AllReduce` operation, but does not move the key and value shards between GPUs. Consequently, in the decoding case, our method benefits from having lower communication volume and suffers less from the communication cost overhead than `Ring Attention` does.

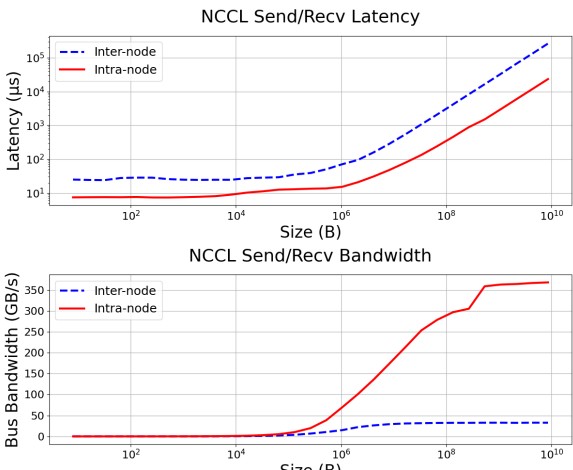

Figure 2: NCCL Send/Recv between two H100 GPUs intra-node and inter-node. GPU clusters offer a two-tier topology where intra-node bandwidth is significantly higher than inter-node. Algorithms such as `Tree Attention` exploit this topology by reducing inter-node communication requirements, enabling better overlap of communication with computation.

**Implications of network bandwidth heirarchy** `Ring Attention` is inherently not topology-aware, and only scales within a network of homogeneous bandwidth. However, this is in conflict with the two-level network topology of modern GPU clusters, which use high-bandwidth interconnects within nodes (NVLINK or PCIe) and comparatively lower-bandwidth interconnects across nodes (InfiniBand or Ethernet). The interconnects greatly differ in bandwidth and latency (see Figure 2). Therefore, `Ring Attention` is bottlenecked by the slowest interconnect, and cannot always overlap the attention computation with communication. We discuss this point further in 6.3 `Tree Attention` improves on `Ring`

`Attention` by using network topology-aware communication patterns to increase overlap of computation and communication, and decrease this scalability bottleneck on communication from the distributed attention computation.

In practice, collective communication libraries like NCCL attempt to automatically detect what the right communication strategy is based on considerations such as data volume and network topology. In DGX clusters, for collective operations within a node, ring reduce is performed whereas a tree reduction is performed across nodes. We see that therefore using built-in collective operations such as `Allreduce` leads to a better performance when decoding from long contexts across multiple GPUs than enforcing the `Ring Attention`'s point to point communication pattern. We show how the following strategy outperforms `Ring Attention` when decoding from very long contexts across multiple GPUs.

In our empirical experiments , we use `Flash Attention 2` (Dao, 2023) within each device, both for our algorithm and for `Ring Attention`[2]. We provide a simple JAX implementation of our method in Appendix D. Note that our method mirrors `Flash Decoding` (fla, 2024) except in that case, the parallelization happens at the level of different streaming multiprocessors (SMs) within a GPU whereas we parallelize between different GPUs. All computations are performed in BF16.

## 6 RESULTS

Similar to `Ring Attention`, `Tree Attention` is an exact computation of attention. Since training and evaluation metrics are the same as for attention, our experimental results are focused primarily on latency in section 6.1, peak memory usage in section 6.2 and communication volumes in section 6.3. Since our algorithm computes numerically identical results as the forward pass of standard attention, our performance results transfer seamlessly to transformer architectures.

We performed experiments in Sections 6.1 to 6.3 on a DGX H100 cluster consisting of 16 nodes, each containing 8 H100 GPUs. All GPUs within the node are connected via an all-to-all NVLINK 4.0 (900GBps) topology. Nodes are connected to each other via 8 InfiniBand NDR interconnects per node (1 per GPU), each of which provides 400 Gbps (leading to an aggregate 3.2 Tbps node injection bandwidth).

We also show `Ring Attention` and `Tree Attention` comparisons when used in a Llama 3 model (Grattafiori et al., 2024) in Sections 6.4 and C.3 on viarous GPU and interconnect types: 8 H100 GPUs with NVLINK 4.0, 8 AMD MI300X GPUs with AMD infinity fabric for intra-node communication and RoCE for inter-node communication, and 2 RTX 4090 GPUs with PCIe interconnect.

### 6.1 LATENCY

In terms of practical usefulness, our study of the energy function brought to light a previously unnoted parallelizability inside the attention computation – that of the reduction of the logsumexp across the sequence dimension, which can be implemented as a parallel `Allreduce`. As stated in Theorem 1, it becomes theoretically possible to implement attention, per query as an $N/p + \log(p)$ parallel operations rather than $N$, where the logarithmic term is proportional to the number of devices available for parallelization. When the attention is sharded across multiple devices, this asymptotic speedup creates a considerable speedup over alternative methods for decoding.

To empirically test the theoretical benefits of our `Tree Attention` method, we compute latency by measuring the time required to perform decoding for different sequence lengths and varying number of H100 nodes. We compare `Tree Attention` to our own `Ring Attention` execution times in Fig. 3. Both methods use `Flash Attention 2` Dao (2023) for the individual-GPU attention computation. For our experiments, we benchmark on a standard attention block consisting of 16 heads of dimension 128 across different sequence lengths.

Our latency results shows how `Tree Attention` improves over `Ring Attention` as we increase the sequence length in Fig. 3(a) and increase the number of GPUs in Fig. 3(b). To better highlight execution time trends with an increasing sequence length, we have also added relative

---

[2]A JAX-based `Ring Attention` implementation that uses `Flash Attention 2` can be found here: https://github.com/nshepperd/flash_attn_jax.

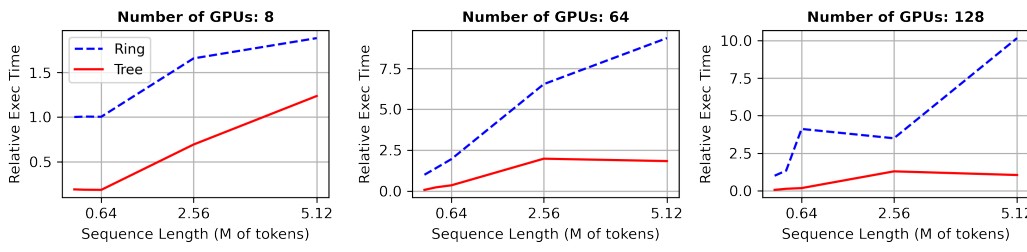

(a) Relative Execution time at different sequence lengths

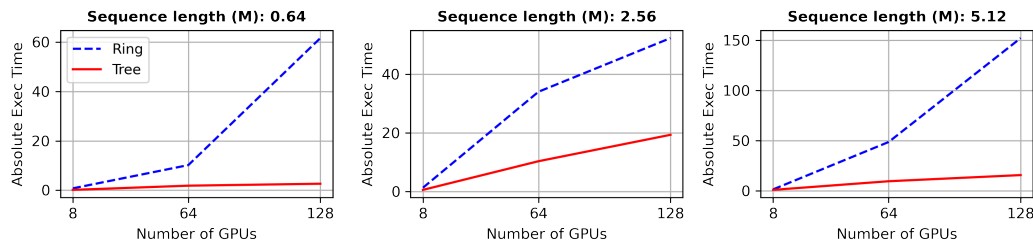

(b) Absolute Execution time for varying cluster sizes

Figure 3: Execution time of 16-head `Tree Attention` vs `Ring Attention` for different sizes of GPU cluster (from 1 to 16 H100 DGX nodes). Relative execution times are indexed to the `Ring Attention` times at a sequence length of 80k tokens.

execution time of both methods with respect to the execution time of ring attention at a sequence length of 80k. With relative execution time in Fig. 3(a), we notice that Tree attention's execution time flattens as the number of GPUs increases, while Ring Attention relative execution time continues to increase. As the plots demonstrate, as we scale the sequence length or the number of GPUs, the gap between `Tree Attention` and `Ring Attention` execution time widens *asymptotically*. Remarkably, `Tree Attention` achieves close ×8 speedups when we use 128 GPUs on a sequence length of 5.12M. We expect this trend to continue for larger sequence lengths. Please note that our DGX cluster is made up of 16 nodes each with 8 GPUs. Results for 8 GPUS use one node, for 64 GPUs uses 8 nodes and for 128 GPUs uses 16 nodes.

## 6.2 MEMORY COST

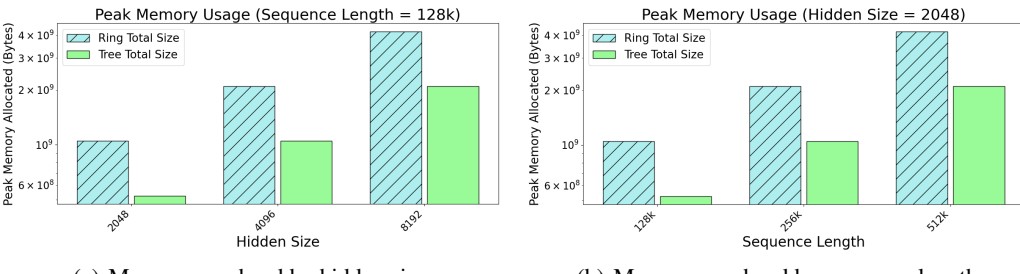

(a) Memory overhead by hidden size          (b) Memory overhead by sequence length

Figure 4: Peak memory usage of a single attention block with `Tree Attention` vs `Ring Attention` when sharded between two RTX 4090s. Results were taken using the JAX memory profiler on one GPU. The difference in peak memory scales with hidden size and sequence length.

To perform `Ring Attention` with a distributed KV cache, it is necessary to broadcast the query corresponding to the final element of the sequence back to all devices, as outlined in step 2 of our Algorithm 1. Each device will then hold a tuple $(\mathbf{q}, \mathbf{k}_{\hat{a}}, \mathbf{v}_{\hat{a}})$, where $\hat{a}$ is the chunk index, which includes the query vector and a local chunk of the keys and values specific to the sequence chunk on that device. The memory cost to store these objects is the same as for `Tree Decoding`.

Additionally, `Ring Attention` must store the $\mathbf{k}_{\hat{a}'}$, $\mathbf{v}_{\hat{a}'}$ coming from the neighbouring device and the chunk of the output $o$ that has the same shape as the query held by that device. In contrast, our method requires storing the communicated chunk of the numerator n, denominator d and max $m$. We do not pre-allocate an output tensor but instead just return the result of doing the `Allreduce` to the numerator divided by the Allreduced denominator. In summary we have the following peak memory costs for Ring and Tree attention:

$$\text{Mem}_{\text{ring}} \quad = 4btd + 2bd \tag{8}$$
$$\text{Mem}_{\text{tree}} \quad = 2btd + 2bd + 2bn_h, \tag{9}$$

where $d = d_h \times n_h$, for head size $d_h$ and $n_h$ number of heads, $b$ denotes the batch size and $t = N/p$. As such, so long as $2bn_h \leq 2btd$, which will almost always be the case in realistic scenarios, our method always has a lower peak memory cost compared to `Ring Attention`.

We empirically measure peak memory utilization for our approach and `Ring Attention` to show that indeed memory usage is significantly less for `Tree Attention` in Figure 4. As predicted by theory, scaling hidden size or sequence length scales `Ring Attention` peak memory usage about 2× faster than `Tree Attention`. For example, doubling the hidden size from 2048 to 4096, doubles the gap in peak memory between two methods, going from 524MB to 1040MB.

## 6.3 COMMUNICATION VOLUME

For `Ring Attention`'s P2P communication strategy, the total volume of data being communicated between devices (in units of number of tensor elements) per iteration scales with $p$ and is given by:

$$V_{ring} = 2btd \times p \tag{10}$$

where $p$ is the number of devices. The first factor comes from counting the total number of communicated elements corresponding to $\{(\mathbf{k}_{\hat{a}}, \mathbf{v}_{\hat{a}}), \hat{a} = 1, \cdots, t\}$, i.e.

$$\text{numel}\left(\{(\mathbf{k}_{\hat{a}}, \mathbf{v}_{\hat{a}}), \hat{a} = 1, \cdots, t\}\right) = 2btd. \tag{11}$$

The `Allreduce` strategy we use in `Tree Decoding` requires the following volume Anthony et al. (2024):

$$V_{\texttt{Allreduce}} = 2 \times \frac{p-1}{p} \times \text{numel}. \tag{12}$$

We communicate a shard of the numerator, denominator and max, requiring:

$$\text{numel}\,(\text{n}, \text{d}, m) = bd + 2bn_h. \tag{13}$$

Note that we first perform on device the local reductions to obtain the local numerator and denominator on each device which consequently makes it so that $t$, i.e. the size of the local sequence chunk does not appear in the above expression. We then obtain:

$$V_{Tree} = 2\frac{p-1}{p} \times (bd + 2bn_h)\,. \tag{14}$$

Our theoretical analysis shows that per iteration our algorithm maintains a lower communication volume than `Ring Attention`. Note however that `Ring Attention` when performed in the training setting with many queries overlaps communication and computation so as to hide its communication costs. However, overlapping communication and computation in the decoding case is infeasible because of how fast the attention computation on a single GPU is relative to how long it takes to communicate the chunk of keys and values between two devices.

Concretely, let us take the example of decoding from a context of length 640000 split between 8 GPUs within one node. Let us take a hidden size of 2048 and fix our data type to be `bfloat16`. Each device for decoding takes $O(10^{-5})$ seconds to perform the `Flash Attention` computation. The time it takes to move the keys and values of the corresponding size between adjacent GPUs as per Fig. 2 is roughly $O(10^{-3})$ seconds. The latency incurred between nodes is even greater and therefore overlapping is not feasible due to this disparity in timescales.

## 6.4 PERFORMANCE WITH A LLAMA TRANSFORMER MODEL

To show that `Tree attention` can also be used in real world applications, we also measured end-to-end throughput with the Llama 3.1 8B model Grattafiori et al. (2024) on prompt sequences of length 32k, 64k, 128k and 256k using ring attention or tree attention for decoding (with prefill) 10 tokens in Table 1. We ran these experiments on 8 H100 GPUs in a DGX cluster (connected with NVLink) as well as 4 MI300X GPUs in an AMD cluster connected with AMD infinity fabric. In Table 2 of Appendix C.3, we also show similar throughput results on 2 RTX 4090 GPUs connected with PCIe. In all cases we see that `Tree attention` for decoding has significantly lower latency than `Ring Attention` for decoding with a prefill stage. `Ring Attention` is up to ×4 faster using 8x H100s and up to ×3 faster using 4x MI300x. We expect this gap to increase as we increase the number of nodes.

While we have previously discussed that `Ring Attention` works best when used with the Ring Topology of TPU clusters, Table 1 and 2 show that `Tree Attention` results generalize well to various types of systems, number of GPUs, communication protocols and network topologies.

Table 1: Average Decoding Time with a prefill stage (in seconds) comparisons, using the 8B Llama 3.1 model with `Tree Attention` (ours) and `Ring Attention` (SOTA) across various sequence lengths and GPU types. Average results and standard error (±) are computed using 10 trial runs.

| Sequence Length | 8x H100s | | | 4x MI300x | | |
|---|---|---|---|---|---|---|
| | Tree Attn | Ring Attn | Speedup | Tree Attn | Ring Attn | Speedup |
| 32k | **0.60** ± 0.15 | 2.57 ± 0.35 | ×4 | **1.05** ± 0.01 | 3.57 ± 0.25 | ×3 |
| 64k | **1.08** ± 0.10 | 4.42 ± 0.38 | ×4 | **2.36** ± 0.01 | 7.33 ± 0.25 | ×3 |
| 128k | **2.68** ± 0.28 | 6.38 ± 0.58 | ×2 | **6.43** ± 0.25 | 16.40 ± 0.40 | ×3 |
| 256k | **2.89** ± 0.62 | 8.19 ± 1.07 | ×3 | **15.30** ± 4.93 | 35.12 ± 5.02 | ×2 |

## 7 DISCUSSION AND CONCLUSION

In this paper, we have derived the energy function for self-attention and demonstrated how the computation of the derivative of this function provides a novel and efficient method for computing attention in parallel. This advantage is especially apparent when performing decoding across multiple devices, in which case our `Tree Attention` enables us to substantially outperform SOTA `Ring Attention` with an *asymptotically* superior algorithm, with ×8 speedups when we use 128 GPUs on a sequence length of 5.12M. We also see that the `AllReduce` operation that we use involves sending partially reduced objects, which greatly reduces the volume of communicated data as well as the peak memory requirement. In a real-world application, using the Llama 3.1 model with 1B and 8B parameters, we find that decoding with a prefill stage using `Tree Attention` gets us ×3-5 speedupds compared to `Ring Attention`. Further, by testing our method on various types of GPUs clusters including AMD MI300xs, we show that `Tree Attention` generalizes very well to various communication protocols and network topologies.

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

## A    MORE RELATED WORK

Recent work has attempted a Bayesian reformulation of attention by deriving a probabilistic generative model which matches the operations performed in a self-attention operation Singh & Buckley (2023). This follows on from a line of work which relates self-attention to the well-studied Hopfield Network architecture Ramsauer et al. (2020). The idea being that while the Hopfield Network is defined in terms of dynamics upon an energy landscape, this same picture can be cast into a Bayesian interpretation by identifying the energy function with a variational free energy functional and hence deriving the generative model that is implicit in the self-attention update.

In particular, consider the energy function proposed in Hoover et al. (2023), which is the logsumexp. Since the gradient in the update rule of that paper is taken with respect to the input to the block, the resulting function is a modified version of the self-attention operation. Similarly, the update rule in Ramsauer et al. (2020) requires the tying of certain weights (K and V) in the attention operation. This restricts the Hopfield derivation to modelling auto-associative retrieval networks, while transformer attention is hetero-associative.

Another notable related work is Feng et al. (2024), where the authors made similar observations as we do in section 5 about how the associative operations within the attention computation can be efficiently parallelized to motivate an attention-based modified RNN architecture for sequence modeling.

While this energy function by itself is primarily a mathematical and theoretical curiosity, we demonstrate below that when combined with automatic differentiation, our formulation naturally leads to highly efficient parallel algorithms for computing attention and performing decoding, especially across multiple devices.

## B    MORE BACKGROUND ON THE TREE REDUCTION OPERATION

A tree reduction operation is a hierarchical strategy to perform a reduction operation (e.g., sum, product, maximum, minimum) over a set of data elements efficiently, especially in parallel computing. This approach reduces the overall computational complexity and enables efficient utilization of parallel processing resources. Here's how it works:

- Divide the problem into smaller tasks: The input data is divided into smaller chunks, and the reduction operation is performed pairwise between adjacent elements in these chunks.

- Form a tree-like structure: The results from the first level of reductions are themselves reduced pairwise in the next level. This continues until the entire dataset has been reduced to a single result.

- Iterative or recursive aggregation: The aggregation typically follows a binary tree pattern, but other fan-in numbers (e.g., k-ary trees) can also be used. Each node in the tree represents a partial reduction result, and the root of the tree holds the final result.

Because a tree structure has a logarithmic depth to total number of nodes, a tree reduction can asymptotiacally reduce the number of total steps required to perform an operation when it is possible to aggregate partial results, and additionally is amenable to parallelization since k-ary trees can be defined to match the number of available processors for parallel processing. Additionally, many existing networking topologies such as Nvidia's NVLINK and Infiniband, due to the natural advantages of tree structures, are designed with such a toplogy meaning that tree operations are natural and efficient to perform.

## C    ATTENTION AS THE GRADIENT OF AN ENERGY FUNCTION AND BAYESIAN INTERPRETATIONS

### C.1    PROOF OF OBSERVATION 1

Here, we show how the self-attention operation can be written as the gradient of an energy function. In particular, we define a scalar function that depends on the keys, queries, values and additionally on an auxiliary vector that we refer to as the *source* $\zeta$. The source is the parameter with respect to which

we compute the gradient of the scalar function to obtain the self-attention operation. We need the source in order to write down the generating function of the moments of the distribution above. It is also the variable with respect to which we can Taylor-expand the generating function and extract the moments as the coefficients of the monomials of $\zeta$ appearing in the Taylor series. Explicitly, we want to find a function $F(q, k, v, \zeta)$ such that:

$$\sum_{a=1}^{N} \text{softmax}(q \cdot k_a) v_a = \left. \frac{\partial F}{\partial \zeta} \right|_{\zeta=0}. \tag{15}$$

This terminology is inspired by work on energy-based models in machine learning Beal (2003); LeCun et al. (2006); Song & Kingma (2021). A summary of variables and indices is provided in appendix G

We first show how the energy function is given by the cumulant-generating function associated to the distribution given by attention scores. Taking inspiration from statistical mechanics, where an analogous cumulant-generating function defines the Helmholtz Free energy (Landau & Lifshitz, 1958), we dub our cumulant-generating function the *energy function for self-attention*.

Let us focus on the case with a single query. As noted above, we leverage the fact that the attention operation can be seen as the computation of the expectation value of the vectors $v$ in the distribution set by the attention scores $z$:

$$z = \langle v \rangle = \sum_{a=1}^{N} P_a v_a = \frac{\sum_{a=1}^{N} e^{q \cdot k_a^T} v_a}{\sum_{i=1}^{N} e^{q \cdot k_i^T}}. \tag{16}$$

The probability density is given by:

$$P_a = \frac{e^{q \cdot k_a^T}}{\sum_{i=1}^{N} e^{q \cdot k_i^T}}. \tag{17}$$

Typically, the denominator or normalization factor is identified with the so-called partition function:

$$Z = \sum_{a=1}^{N} e^{q \cdot k_a^T}. \tag{18}$$

We can now compute the first moment of the probability distribution given above by introducing a source, $\zeta \in \mathbb{R}^d$. In our case, with $\zeta$, we can extend the partition function to the function:

$$Z(\zeta) = \sum_{a=1}^{N} e^{q \cdot k_a^T + \zeta \cdot v_a^T}. \tag{19}$$

Now, we can compute any moment of the distribution as the $n$-th Taylor coefficient of $Z(\zeta)$ $\forall A_1, A_2, \cdots \in \{1, \cdots, d_h\}$ :

$$\langle v_{A_1} \cdots v_{A_n} \rangle = \frac{1}{Z} \left. \frac{\partial^n Z(\zeta)}{\partial \zeta_{A_1} \cdots \partial \zeta_{A_n}} \right|_{\zeta=0}. \tag{20}$$

In other words, we can write $Z(\zeta)$ as:

$$Z(\zeta) = Z \left( 1 + \langle v \rangle \zeta + \frac{1}{2!} \langle v_{A_1} v_{A_2} \rangle \zeta_{A_1} \zeta_{A_2} + \cdots \right) \tag{21}$$

Therefore, the first moment can be written as:

$$\langle v \rangle = \frac{1}{Z} \left. \frac{\partial Z}{\partial \zeta} \right|_{\zeta=0}, \tag{22}$$

which can be written as the gradient of the log of $Z(\zeta)$:

$$\langle v \rangle = \left. \frac{\partial}{\partial \zeta} \log Z(\zeta) \right|_{\zeta=0}. \tag{23}$$

This quantity is the generating function, a.k.a. the free energy:

$$F = \log \sum_a \exp\left(q \cdot k_a^T + \zeta \cdot v_a^T\right). \tag{24}$$

To compute causal self-attention, we introduce $N$ sources $\zeta^i$ each $\in \mathbb{R}^d$ and take

$$F_{tot} = \sum_{i=1}^{N} F_i = \sum_{i=1}^{N} \log \sum_{a=1}^{i} \exp(q_i \cdot k_a^T + \zeta_i \cdot v_a^T). \tag{25}$$

The truncation of the inner sum up to index $i$ is due to causal masking.

Now, in order to compute the $i$-th element of causal self-attention, we differentiate with respect to $\zeta_i$ and set it to zero:

$$\left.\frac{\partial F_{tot}}{\partial \zeta_{i,A}}\right|_{\zeta_i=0,\forall i} = \frac{\sum_{a=1}^{i} \exp(q^i \cdot k_a^T)v_{a,A}}{\sum_{a=1}^{i} \exp(q^i \cdot k_a^T)}. \tag{26}$$

The generalization to the multi-head attention case is straightforward. In this case, there is one key, query and value per head. For $n_h$ total heads, the generating function takes the form:

$$F_{tot} = \sum_{i=1}^{N} \sum_{h=1}^{n_h} F^{i,h}, \tag{27}$$

where

$$F_{i,h} = \log \sum_{a=1}^{i} \exp\left(q_{i,h} \cdot k_{h,a}^T + \zeta_{h,i} \cdot v_{h,a}^T\right). \tag{28}$$

The output projection weight is included in the definition of $v_j$ here, meaning that

$$v_{b,A} = x_{b,\bar{B}}(W_O W_V)_{A,\bar{B}} \tag{29}$$

where $W_O \in \mathbb{R}^{d_h} \times \mathbb{R}^{d_{emb}}$ denotes a head size slice of the output projection weight and $\bar{B} \in \{1, \cdots, d_h\}$ spans the intra-head indices. In the index notation above, the head indices are barred whereas the embedding space indices are unbarred. We proceed focusing on the single-head case, as it makes the presentation simpler, and the multi-head generalization is immediate. Note that we demonstrate that our energy function approach also can account for safe softmax in Appendix F

## C.2 BAYESIAN INTERPRETATION

The fact that it is possible to derive the self-attention operation as the minimization of an energy function implies that it is possible to provide a Bayesian gloss on self-attention by identifying a likelihood function and showing that we can obtain the forward pass of the attention block from computing the maximum a posteriori estimate of this likelihood.

In particular, we propose the following for the log-likelihood function:

$$\Gamma(\zeta, z) = \sum_{i=1}^{N} \sum_{A=1}^{d} \left(z_{i,A}\zeta_{i,A} - F(\zeta, x)\right). \tag{30}$$

We denote by $x$ the input to the self-attention block from which we obtain $q, k, v$ from multiplying it by the weights $W_Q, W_K, W_V$ respectively. Let us minimize the above with respect to $\zeta$ and $z$ simultaneously:

$$\frac{\partial \Gamma}{\partial \zeta_{i,A}} = 0, \frac{\partial \Gamma}{\partial z_{i,A}} = 0. \tag{31}$$

These conditions written explicitly read

$$\zeta_{i,A*} = 0, \quad z_{i,A*} = \frac{\partial F}{\partial \zeta_{i,A}}. \tag{32}$$

Plugging the first condition into the second leads to the attention forward pass:

$$z_{i*,A} = \frac{\sum_{a=1}^{i} e^{q_i \cdot k_a^T} v_{a,A}}{\sum_{b=1}^{i} e^{q_i \cdot k_b^T}}. \tag{33}$$

In all, this means we can obtain the gradient w.r.t. $\zeta$ from MAP estimation of the following likelihood:

$$z_{i*,A}, \zeta_{i*,A} = \text{argmax}_{\zeta,z} e^{-\Gamma(\zeta,z)}. \tag{34}$$

Moreover, such a procedure enables us to identify the energy-based model associated with the self-attention function.

### C.3 MORE PERFORMANCES RESULTS WITH A LLAMA TRANSFORMER MODEL

To extend our work in section 6.4, and to demonstrate that Tree Attention can be successfully applied to a range of hardware setups, we also experiment with running Llama3.2-1B on a dual NVIDIA RTX 4090 setup. The two 4090s are connected via PCIe networking. Even in this case, we observe a significant 4x speedup (growing to 5x at longer sequence lengths) of Tree Attention over Ring Attention for autoregressive decoding.

Table 2: Average Decoding Time (in seconds) comparisons with prefill stage using the 1B Llama 3.2 model with `Tree Attention` (ours) and `Ring Attention` (SOTA) across various sequence lengths for 4090s. Average results and standard error ($\pm$) are computed using 10 trial runs.

| Sequence Length | Tree Attention | Ring Attention | Speedup |
|:---:|:---:|:---:|:---:|
| | Time (s) | Time (s) | |
| 8000 | $0.34 \pm 0.05$ | $1.38 \pm 0.07$ | $\times 4$ |
| 16000 | $0.58 \pm 0.07$ | $2.77 \pm 0.04$ | $\times 5$ |
| 20000 | $0.74 \pm 0.01$ | $3.47 \pm 0.04$ | $\times 5$ |
| 32000 | $1.01 \pm 0.02$ | $5.45 \pm 0.03$ | $\times 5$ |

## D APPENDIX: JAX CODE

Below is the *tree_flash_decode* method. Our full code base is available here: `https://anonymous.4open.science/r/tree_attention-7C32`.

```python
import jax
from jax import lax
import jax.numpy as jnp
from functools import partial
from jax.sharding import Mesh,NamedSharding, PartitionSpec as P
from jax.experimental import mesh_utils
from jax.experimental.shard_map import shard_map
from flash_attn_jax.flash import _flash_mha_vjp

in_specs=(P(None, None, None, None), P(None, 'i', None, None), P(None,
    'i', None, None))
out_specs=P(None, None, None)

@jax.jit
@partial(shard_map, mesh=mesh, in_specs=in_specs, out_specs=out_specs,
    check_rep=False)
def tree_flash_decode(q, k, v):
  def flash_num_lse(q, k, v, config=dict(softmax_scale=1.0,
      is_causal=False, window_size=(-1, -1))):
    tup = _flash_mha_vjp.fwd(q, k, v, config)

    res,lse = tup[1][3],tup[1][4]
    return res,lse
```

```
loc_res, loc_lse = flash_num_lse(q, k, v)
a_max_global = lax.pmax(loc_lse, axis_name='i')

num_global = lax.psum(loc_res * jnp.exp(loc_lse - a_max_global),
    axis_name='i')

den_global = lax.psum(jnp.exp(loc_lse - a_max_global), axis_name='i')

return (num_global / den_global)
```

The function uses Flash Attention 2 Dao (2023) to compute the local numerator and denominator, both of which are accumulated between devices using an `Allreduce` (which is what psum and pmax call). NCCL determines in what pattern these results are communicated.

## E  THEOREM 1 PROOF

We prove theorem 1 below.

*Proof.*

**Sequential Case:** On a single GPU, the reduction operation over an array of size $N$ has a time complexity of $O(N)$ since the processor must sequentially process each element.

**Parallel Processing with $p$ Processors:** Divide the array of size $N$ into $p$ chunks, each of size $\frac{N}{p}$. Each processor performs the reduction operation on its chunk independently. The time complexity for each processor is $O\left(\frac{N}{p}\right)$.

**Combining Partial Results:** The partial results from the $p$ processors need to be combined. Using a tree pattern for reduction, the partial results can be reduced in $O(\log p)$ steps. Each step involves combining pairs of results, halving the number of results at each step until only one result remains.

**Total Time Complexity:** The total time complexity is the sum of the time complexities for processing the chunks and combining the results:

$$O\left(\frac{N}{p}\right) + O(\log p).$$

This proves that the time complexity of a reduction involving an associative operation over an array of size $N$ is $O\left(\frac{N}{p} + \log p\right)$ when using $p$ parallel processors, and it reduces to $O(\log N)$ when the number of processors is equal to the size of the array. $\qquad\square$

## F  COMPUTING SAFE SOFTMAX

While, mathematically, attention utilizes the softmax operation, in practice this is often numerically unstable using relatively low precision operations. To address this, a mathematically equivalent function, the 'safe softmax' is instead used which subtracts all dot products in the exponential by the max. This ensures that all values being exponentiated are less than 1 and hence less likely to explode and cause numerical instability. Here, we demonstrate that our energy function approach also can account for safe softmax.

Let us suppose we compare our generating function

$$F_{tot} = \sum_i \log \sum_{a=1}^{i} \exp\left(q_i \cdot k_a^T + \zeta_a \cdot v_a^T\right) \tag{35}$$

and a slightly modified one:

$$F'_{tot} = \sum_i \log \sum_{a=1}^{i} \exp\left(q_i \cdot k_a^T + \zeta_i \cdot v_a^T - m_i\right). \tag{36}$$

When we take the derivative of these two quantities, we see that we get the same result:

$$\left.\frac{\partial F_{tot}}{\partial \zeta_i}\right|_{\zeta_i=0} = \left.\frac{\partial F'_{tot}}{\partial \zeta_i}\right|_{\zeta_i=0}. \tag{37}$$

To see it explicitly:

$$\left.\frac{\partial F'_{tot}}{\partial \zeta_i}\right|_{\zeta_i=0} = \frac{\sum_{a=1}^{i} \exp(q_i \cdot k_a^T - m_i)v_a}{\sum_{a=1}^{i} \exp(q_i \cdot k_a^T - m_i)} \tag{38}$$

$$= \frac{\sum_{a=1}^{i} \exp(q_i \cdot k_a^T)v_a}{\sum_{a=1}^{i} \exp(q_i \cdot k_a^T)}.$$

Normally, when computing the softmax in an online fashion, this procedure is performed where $m_i$ is the row max of $q \cdot k^T$. This shift makes it so that the sum of exponentials doesn't lead to overflows.

## G    NOTATIONS FOR EQUATIONS

Here is a summary of the various variables and indices that will be used in the coming sections:

TABLE I: Variable names.

| | |
|---|---|
| $x$ | Attention Input |
| $q, k, v$ | Query, key and value vectors |
| $\Gamma$ | Attention Log-likelihood |
| $\zeta$ | Source vector |
| $m$ | Max of $q \cdot k^T$ |
| $Z$ | Partition function |
| $z$ | Activation vector |
| n | Attention numerator |
| d | Attention denominator |
| lse | Attention score logsumexp |
| $F$ | Generating function |
| $P$ | Attention score probability density |

$$\tag{39}$$

TABLE II: Index names and ranges.

| | |
|---|---|
| $N$ | Sequence length |
| $d$ | Embedding dimension |
| $d_h$ | Head dimension |
| $p$ | Number of devices |
| $t$ | Chunk size N/p |
| $b$ | Batch size |
| $a, i, j \in \{1, \cdots, N\}$ | Sequence Indices |
| $A, B \in \{1, \cdots, d\}$ | Embedding indices |
| $\bar{A}, \bar{B} \in \{1, \cdots, d_h\}$ | Intra-head indices |
| $h \in \{1, \cdots, n_h\}$ | Head indices |
| $\hat{a}, \hat{b} \in \{1, \cdots, t\}$ | Intra chunk indices |

$$\tag{40}$$

