# OpenReview forum: "Tree Attention: Topology-Aware Decoding for Long-Context Attention"
_ICLR.cc/2025/Conference — Submitted to ICLR 2025_

### Official Review · Reviewer_Eqp7 · 2024-11-04

**Soundness:** 2
**Presentation:** 2
**Contribution:** 2
**Rating:** 3
**Confidence:** 3

**Summary:**

This paper proposes a new attention mechanism called tree attention. By leveraging tree reduction, it enables high parallelization of the decoding stage during LLM inference, especially for extremely long contexts.

**Strengths:**

1. This paper includes a microbenchmark of latency and bandwidth between inter-node and intra-node H100 GPUs, highlighting the communication bottleneck.

2. The paper introduces a new approach to designing improved attention mechanisms by considering the system topologies.

**Weaknesses:**

1. Although this work is called "tree attention," it is not general enough to handle common attention operations. It can only be used when the query has a single token, which limits its usefulness during training and the prefill stage.

2. The implementation of the ring attention baseline might be incorrect. The authors directly use the ring_fwd function from the flash_attn_jax library, which assumes the query, key, and value matrices are already partitioned [1]. Consequently, the actual calculation in this paper's implementation [2] involves a query matrix with P tokens, where P represents the number of workers.

3. Ring attention is not an appropriate baseline for the scenario considered in this work. The purpose of ring attention is to address long-context training, where the query, key, and value matrices are partitioned across workers. To calculate the complete attention map, these matrices need to be communicated, and ring attention provides a better scheduling mechanism to overlap communication and computation. However, in the scenario discussed here—during the decoding stage—the query has only one token and is copied to all workers. Therefore, only the last row of the attention map needs to be calculated, making ring attention unnecessary.

4. The idea and algorithm presented are straightforward, but the authors appear to complicate the explanation, which adds confusion rather than clarity.

5. The evaluation only demonstrates a single attention module and lacks end-to-end evaluation on a real-world large language model (LLM), which could weaken the authors’ claims about the algorithm’s effectiveness.

6. The algorithm and code do not utilize a tree topology. If they merely rely on the NCCL library, this approach may lack generalizability, reducing this work's contribution.

[1] https://github.com/nshepperd/flash_attn_jax/blob/eab32ee0725f1de66467f35ecd7f6acd15dd3f1f/tests/test_ring.py#L76

[2] https://anonymous.4open.science/r/tree_attention-7C32/ring_shard_test.py

**Questions:**

Please address the issues mentioned in the weaknesses section.

---

> ### Comment · Reviewer_Eqp7 · 2024-11-28
>
> As the authors have not addressed the reviews, I will maintain my initial rating. The paper has several significant issues and is not well-prepared in its current form. Therefore, I believe it is not suitable for acceptance.

---

> > ### Author Response · Authors · 2024-12-01
> >
> > We apologise for the delay in our response. We note that the author-reviewer discussion period has been extended to December 2nd. It would be great if you could consider our recent response and updates to the manuscript.

---

> > > ### Author Response · Authors · 2024-12-02
> > > **Reviewer Eqp7: we have one more day to discuss Tree Attention**
> > >
> > > Dear Reviewer Eqp7:
> > >
> > > The discussion period has not yet ended. We believe to have addressed all of the points that you have raised. Please take into consideration the work that we have done to improve our story line and to show the impact of Tree Attention's strong performance for:
> > > 1. a variety of GPU types,
> > > 2. in real-world models such as Llama and
> > > 3. compared against SOTA multi-GPU attention parallelization techniques such as Ring Attention.
> > >
> > > Thank you.

---

> ### Author Response · Authors · 2024-11-29
> **Answer to Reviewer Eqp7**
>
> We appreciate your review and feedback and agree that Tree Attention proposes a novel formulation of Attention that enables lower memory, lower communication volume and higher throughput. We have improved our paper with some of your suggestions.
>
> Before we address your specific concerns, we wanted to reiterate that Tree Attention is a method to parallelize attention during decoding. We openly state that Ring Attention is able to hide many of its limitations that we have highlighted (see analysis in section 6) by overlapping communication with compute during training (see lines 483 where this is discussed). Further, to the best of our knowledge, there are no other highly multi-GPU LLM decoding schemes apart from *Ring Attention*, which is why *Tree Attention* can be of interest to speed up LLM generation (we have made this last point clearer with an improved Related Works in section 2 taken from the appendix).
>
> We hope to now discuss the mentioned improvement areas of our paper with you:
>
> 1. Tree attention only for decoding (and prefill): As mentioned above, we propose a new formulation of attention to address a need for decoding at scale and for large sequence lengths. Since autoregressive generation forms a very significant part of realistic LLM inference workloads, speeding up the decoding phase alone is still of great interest and importance.
> However, we agree that we should have also shown that *Tree Attention* performs well even when a prefill stage is needed. In table 1 in section 6.4 our results with a **Llama 3** model compares *Tree Attention* and *Ring Attention*. Since the benefits of Tree Attention stack as we stack more layers, we observe an increasing benefit of using *Tree Attention* over *Ring Attention* with **x4 improved speed of decoding with prefill state**. *Tree Attention* for the training regime is also an interesting direction and is left to future work.
>
>
> 2. Ring Attention implementation: We maintain that our implementation of ring attention is correct. We compute the output by broadcasting the query from the final rank to all other ranks, hence we have indeed $P$ query tokens, and computing the output replicated on every rank. This replication doesn’t change the number of ring passes and consequently doesn’t impact the execution time as the communication and computation are overlapped in the ring attention computation. We are happy to discuss further if need be.
>
> 3. Ring Attention as a baseline: Ring attention and Tree attention are methods that sequence parallelize the attention computation across multiple GPUs (see Related Works section 2 where we clarify this point). Even for a query size of 1, this is still the case where the $KV$ computation is sequence parallelized. For long sequences, per device memory limitations are also important and both Tree and Ring attention are a similar class of algorithm that address this issue. Further, in the case where we are decoding and hence have only a single query but many keys and values, we can either:
> a. communicate the query and then the partially computed results of the attention operation and then recombine the results across devices (i.e. Tree Attention).
> b. communicate shards of the keys and values and hold a copy of the query on each device (i.e. Ring Attention).
> *Ring Attention* is the current **SOTA** method for multi-GPU parallelization of Attention even in the case of decoding. Our proposed Tree  Attention method improves on some of the limitations of *Ring Attention*, for decoding with a prefill stage.
>
> 4. On simplifying the explanation: Thank you for this suggestion. We have simplified the explanation to be much more to the point on how efficiencies are gained with Tree Attention (See sections 4 and 5).
>
> 5. On real world applications: This is again a good point and we have added table 1 in section 6.4 of the new version of the paper (also copied above) to show that Tree Attention benefits stack as we stack more transformer attention layers in **Llama 3** model.
>
> 6. On using a general tree topology: The results in table 1 in Section 6.4 show results for a **wide variety of GPU types** and communication libraries. We use *8 H100 GPUs with NVLINK 4.0, 8 AMD MI300X GPUs with AMD infinity fabric for intra-node communication and RoCE for inter-node communication, and 2 RTX 4090 GPUs with PCIe interconnect*. We also would like to remind that Ring Attention works best when used with the Ring Topology of TPU clusters as mentioned in lines 498-500 of the paper. Tree Attention is much more generalizable than Ring Attention since it leverages a wide variety of Tree communication protocols, found in most GPU systems (AMD, Intel, Nvidia).
>
> Thank you in advance for reading this rebuttal and looking forward to discussing further with you.

---

> ### Comment · Reviewer_Eqp7 · 2024-12-02
>
> I thank the authors for the responses. However, I want to elaborate on why your Ring Attention implementation doesn't make sense to me and why I believe Ring Attention is not a suitable baseline. Suppose the K matrix is partitioned into K_1, K_2, K_3, K_4, and the V matrix is partitioned into V_1, V_2, V_3, V_4, with both distributed across four GPUs. The Q matrix contains only one token and is the same across all GPUs.
> In the first iteration, GPU1 calculates the partial result using K_1 and V_1 and then sends the partition to the next worker GPU2. In the next iteration, GPU2 actually performs the same computation that GPU1 already done in the previous iteration. This results in significant redundant computation and unnecessary communication.
> I suggest the authors conduct a thorough survey of how mainstream LLM inference frameworks handle long-context generation and compare their approach carefully with these methods.
>
> Regarding the new experiments, several details and issues remain unresolved:
>
> 1.	What inference framework are you using? Performance can vary widely across different frameworks.
>
> 2.	In this experiment, the prompt length is long (up to 256k tokens), but the generation length is short (10 tokens). It seems most of the workload is in the prefill stage rather than the decoding stage. Can you explain how Tree Attention is applied during the prefill stage or how you handle the prefill stage without Tree Attention?
>
> 3.	The reported performance seems extremely low. In the vLLM benchmark, the LLaMA 3.1 8B model with a 128k context length achieves thousands of tokens per second on a single H100 GPU [1]. In contrast, your setup delivers less than 1 token per second on 8 GPUs. Can you provide references to validate the correctness of your measurements?
>
> 4.	The time complexity looks different from your earlier analysis.
>
> Furthermore, what is the difference between ring and tree topologies when only two GPUs are used? Based on the experiment with two 4090 GPUs, it appears that the claimed improvement has little to do with the tree topology.
>
> [1] https://github.com/LambdaLabsML/vllm-benchmark/blob/main/benchmark_v0.md

---

> ### Author Response · Authors · 2024-12-03
> **On baselines: Ring and Tree Attention are Sequence Parallel Attention methods; other long-context decoding such as Tensor or Context Parallelization are orthogonal methods**
>
> Thank you very much for continuing the discussion. We really appreciate your input since it will help us clarify certain points in the article. We first continue our discussion on Ring Attention as a baseline below and will address numbered concerns in another comment.
>
> We have emphasized in the article that Tree Attention is a **Sequence Parallel Attention Method**. As we mention in lines 116-117 of the *Related Works*, Ring Attention is the only other method that is also a **Sequence Parallel Attention Method**. We will also make sure to add this point to the abstract and we will also add to the Related Works section:
>
> > Other methods, considered orthogonal to **Sequence Parallel Method on Attention**, include **Tensor Parallel** and **Context Parallel** methods [1,2] that can handle long-context generation and can be used in conjunction with Ring Attention and Tree Attention. Since we parallelize exact attention across multiple devices, Ring Attention (Liu et al., 2023) is the most comparable to our work. Ring Attention shards $K, V$ sequences and communicates $K_i, V_i$ chunks across devices to compute attention, allowing the method to sequence parallelize the attention computation for large sequences *without needing tensor or context parallelization*. While sharding $K, V$ sequences is useful to avoid out-of-memory issues, during decoding with a single query that is copied to all GPUs, Ring Attention can *performs redundant* computation which is (1) overlapped with communication and (2) does not add to the execution time.
>
>
> As we will notice, we do mention that, for decoding, Ring Attention incurs *redundant computation* however, without *tensor* or *context* parallelism, we still need to chunk $K, V$ across GPUs and communicate partial $K_i, V_i$ chunks to avoid OOM issues. Communication is therefore *necessary*. Further, the redundant computation *does not add to execution time* since it overlaps with the *necessary* communication. If you know of any other **Sequence Parallel Attention Method** other than Ring Attention, we would be more than happy to use it as a baseline. Please let us know if such method exists.
>
>
> `References:`
>
> [1] SGLang: Efficient Execution of Structured Language Model Programs, Zheng et al., 2024
>
> [2] Efficient Memory Management for Large Language Model Serving with PagedAttention, Kwon et al., 2023

---

> ### Author Response · Authors · 2024-12-03
> **Continuing discussion on other raised concerns**
>
> We appreciate the clear list of concerns that you have raised. We reply below.
>
> 1. What inference framework are you using? Performance can vary widely across different frameworks.
>
> To isolate the affects of Tree Attention vs Ring Attention for decoding, we use a pure PyTorch and Flash Attention implementation (see our anonymous repo https://anonymous.4open.science/r/tree_attention-7C32). Our framework allows us to abstract away methods that may influence the speed or memory of long sequence generation and that would entangle a clear comparison between Ring Attention and Ring Attention.
>
> 2. In this experiment, the prompt length is long (up to 256k tokens), but the generation length is short (10 tokens). It seems most of the workload is in the prefill stage rather than the decoding stage. Can you explain how Tree Attention is applied during the prefill stage or how you handle the prefill stage without Tree Attention?
>
> We remind the Reviewer that we only make claims about Tree Attention on decoding in the paper. Prefill takes at most 46% of the time for a sequence of 256k tokens on an 8B Llama model. For this reason, we used chunked prefill for Llama with Tree Attention. Note that Tree Attention prefill is much faster than chunked prefill and, nonetheless, we still get *x4 improved speed of decoding*. We will add in the paper that for the newest Llama results chunked prefill was used for Tree Attention.
>
> 3. The reported performance seems extremely low. In the vLLM benchmark, the LLaMA 3.1 8B model with a 128k context length achieves thousands of tokens per second on a single H100 GPU [1]. In contrast, your setup delivers less than 1 token per second on 8 GPUs. Can you provide references to validate the correctness of your measurements?
>
> As discussed in point 1 above, the pure PyTorch and flash attention allows us to abstract away methods that may influence the speed or memory of long sequence generation. Frameworks such as vLLm on the other hand includes many inference time optimizations that would *not* allow us to straighforwardly compare Ring Attention vs. Tree Attention. However, since many inference time optimizations, PagedAttention and tensor parallelism techniques are orthogonal to our work, we intend to integrate our decoding method with vLLm in the future.
>
> 4. The time complexity looks different from your earlier analysis.
>
> We have not changed anything about the time complexity.
>
> 5. Furthermore, what is the difference between ring and tree topologies when only two GPUs are used? Based on the experiment with two 4090 GPUs, it appears that the claimed improvement has little to do with the tree topology.
>
> By not moving the keys and values but only the queries and partial outputs, we have a much lower communication overhead with Tree Attention even when we are dealing with just two GPUs. "it appears that the claimed improvement has little to do with the tree topology": we mention in the paper that the tree topology is only one element of our overall performance improvements. As we increase the number of GPUs, more communication happens and this is where the tree topology becomes important. In fact, and as proven and discussed a few times in the paper and the rebuttal, when we increase the number of GPUs we see asymptotic improvement in decoding speed.
>
> Thank you.

---

> ### Comment · Reviewer_Eqp7 · 2024-12-03
>
> > Communication is therefore necessary. Further, the redundant computation does not add to execution time since it overlaps with the necessary communication.
>
> Directly reducing the partial results is clearly more efficient than sending chunked KV multiple times in this scenario.
>
> > We have not changed anything about the time complexity.
>
> In table 1, the time complexity on the H100 machine seems to be O(logn), while on MI300X machine, it seems to be O(n), regardless of whether tree attention or ring attention is used. Some explanation or profiling of these results is needed.
>
> > Note that Tree Attention prefill is much faster than chunked prefill.
>
> Chunked prefill is used to better overlap the prefill and decode stages when serving multiple concurrent requests. However, I couldn’t see how you used chunked prefill in the scenario discussed in the paper and the code, or how the comparison makes sense. Please ensure you fully understand the motivation and usage of chunked prefill before making this claim.
>
> > we mention in the paper that the tree topology is only one element of our overall performance improvements
>
> It would be better to explicitly list all the elements and conduct an ablation study to demonstrate the contribution of each component. It is difficult for me to identify the key component of this work that contributes the most to the improvements.

---

### Official Review · Reviewer_mxBy · 2024-11-04

**Soundness:** 2
**Presentation:** 2
**Contribution:** 2
**Rating:** 5
**Confidence:** 2

**Summary:**

This paper addresses the challenge of efficiently computing self-attention in transformer models, which traditionally incurs quadratic complexity in sequence length. The authors propose an approach for parallelizing attention computation using Tree Attention, which leverages the gradient of an energy function. By implementing a tree reduction communication strategy, the proposed approach enables faster parallel decoding and reportedly achieves up to 8x faster performance than Ring Attention, with reduced communication volume and half the memory needed.

**Strengths:**

The Tree Attention approach accounts for two-level network topology awareness within GPU clusters, which is a practical and well-considered enhancement for real-world distributed computing environments.

**Weaknesses:**

- The paper does not clearly explain why using the gradient of the energy function is advantageous for self-attention. This aspect feels insufficiently motivated, making it unclear on the potential contribution. The authors could have strengthened their argument by explaining how this approach uniquely benefits attention computation compared to existing methods.
- The paper claims significant performance improvements over Ring Attention, but it lacks clarity on what specific limitations in Ring Attention are addressed by Tree Attention. Without a clear comparison, it is challenging to understand if Tree Attention fills an existing gap or if it simply provides an alternative approach.
- The details of the Tree Attention approach are challenging to follow, potentially impacting its accessibility and reproducibility. The method relies on parallel decoding and a tree reduction for attention computation, but the details of these operations are not well explained. Clearer explanations or diagrams could be helpful.

**Questions:**

- What design aspects of Tree Attention allow it to remain exact, despite the use of parallelized operations? Does the tree reduction introduce any approximations, or is it truly an exact method? This clarification would be valuable for understanding its applicability in tasks where exact attention computation is critical.
- The paper reports an 8x speedup compared to Ring Attention. Could the authors break down what aspects of Tree Attention contribute to this speedup? For example, how much of the improvement is due to, e.g., reduced communication volume by tree reduction?
- Given that Ring Attention is already a sequence parallel method for multi-GPU attention parallelism, what specific limitations does Tree Attention address that Ring Attention?

---

> ### Author Response · Authors · 2024-11-29
> **Answer to Reviewer mxBy**
>
> Thank you for taking the time to read and review our paper. As you mentioned, Tree Attention was based on a theoretical link between energy functions and attention as seen in Section 4. The Tree Attention algorithm also provides a clear advantage over SOTA attention parallelization methods such as Ring Attention by providing:
>
> 1. lower memory overhead by a factor of 2
> 2. faster throughput (8x faster)
> 3. less communication volume (2x less)
> 4. better scaling with increasing number of GPUs (logarithmic scale)
>
> We address your specific questions below and point to places in the paper where we have expanded on our answers following your feedback.
>
> 1. Regarding the advantage of using the gradient of an energy function for self-attention: We agree that this point was not made clear enough in sections 3 and 4 of the original manuscript. We have re-organized the paper by making Section 3 much more succinct and added further descriptions to highlight the importance of using the energy formulation to describe attention, while moving many of the detailed derivations to an appendix. Our added descriptions are as follows:
>
> > As outlined in Equation 6, the formulation of attention based on the energy function involves the $\log\sum\exp$ operation. This operation is not only parallelizable but can also make use of tree-reduction to lower inter-device communications (Anthony
> et al., 2024), making logsumexp an ideal candidate to parallelize attention across several devices.
>
> This added paragraph will be a link to Sections 5.1 Efficient Energy Function Computation, 5.2 Efficient Parallel Decoding and 5.3 Efficient Collective Operations using Topology-Awareness that describes in detail the exact mechanisms that leverage the energy function in our parallelization of attention.
>
> 2. On the limitations of Ring Attention: Section 6 describes Ring Attention’s algorithm and ring-topology limitations.
> For example:
>
>
> a. In section 5.1. we wrote:
>
> > As we scale the sequence length or the number of GPUs, the gap between Tree Attention and Ring Attention execution time widens asymptotically.
>
> b. In section 5.2, we wrote:
> > Ring Attention must store the k ˆ𝑎′ , v ˆ𝑎′ coming from the neighbouring device, thus increasing the memory requirement.
>
>
> c. In section 5.3, we wrote:
> > Ring Attention’s P2P communication strategy, the total volume of data being communicated between device scales with p.
>
>
>
> However, we think that discussions of Ring Attention limitations should be improved following your feedback. We have added a paragraph that summarizes the points above in the introduction.
>
> 3. Regarding the reproducibility and understandability of our work, we provide explicit algorithms (Section 4), and our code implementation (https://anonymous.4open.science/r/tree_attention-7C32%7D ) in addition to  pseudocode in Appendix C to show how parallelize decoding (using shard_map in Jax for example).
> The tree reduction operation involved in lowering communication costs during the parallel logsumexp computation is described at length in (Anthony et al., 2024). However, following your feedback, we agree that the article should be more self-contained and have added a section on tree reduction in Appendix B.
>
> 4. Is Tree Attention an exact computation of Attention: It is! We show how our formulation using an energy function and the standard attention block are exactly equivalent  in equation 5. Moreover, we reiterate in lines 78-79 that:
>
> > It can be shown empirically that Ring Attention and Tree Attention are exact computations of Attention since both methods have exactly the same activations as the forward pass of Vanilla Attention.
>
> 5. On breaking down speedup contributions, Tree attention leverages a bandwidth/latency tradeoff. We can only examine this tradeoff empirically by varying the number of GPUs and the sequence length and observe the behavior of various attributes of Tree Attention (communication, memory, latency). It is difficult to decouple the factors that lead to speedups in practice (beyond our theoretical analysis already in the paper) because some aspects such as communication buffers are difficult to observe and others such as latency can depend on many idiosyncratic factors of the hardware setup.
>
>
> Thank you.

---

### Official Review · Reviewer_yWiF · 2024-11-04

**Soundness:** 2
**Presentation:** 3
**Contribution:** 2
**Rating:** 5
**Confidence:** 3

**Summary:**

This paper proposes a new method for improving the performance of self-attention mechanisms used in transformers, especially for long-context sequences. The authors introduce a tree-based method for computing attention, which uses the internal parallelism in the attention operation. This method reduces communication overhead and memory usage across multi-GPU clusters, as compared to existing methods like Ring Attention. The paper establishes a theoretical basis for their approach by deriving an energy function for self-attention and showing how it can be efficiently computed using parallel reductions. Experimental results indicate that Tree Attention achieves up to 8× speedups in long-sequence attention tasks while also halving peak memory usage.

**Strengths:**

1. The proposed method can effectively scale with the number of GPUs and scales logarithmically.

2. The memory usage of the proposed method can significantly reduce the memory usage.

3. The paper innovatively improve self-attention mechanism by using other related field's knowledge, which is interesting.

**Weaknesses:**

1. While the paper demonstrates high theoretical and experimental results, it did not provide any real-world deployment or compare with existing frameworks.

2. The system only tests on DGX H100 system, limiting the generalizability to other systems.

3. The paper only compare with self-implemented ring attention, it would be beneficial to compare with more other attention mechanisms like Linformer, Longformer, or performer.

**Questions:**

Overall, I think this paper proposed a novel self-attention mechanism that scales logarithmically and reduces memory footprints. I have a few comments and questions as follow.

In section 3, the derivation of energy function for self-attention is interesting. What will be comparative analysis between tree attention's use of energy function and other attention mechanism would use this energy function?

As stated in the cons, the paper showed an impressive result compare to ring-based attention. What would be the testing results compare to other different attention mechanism with more aspects like (different hidden sizes, varying GPU numbers)? Also, what will be the memory usage compare to other methods?

For large distributed clusters (cloud-based GPUs), communication latency is a major concern, and overlapping communication and computation can be tricky. More insights on how Tree Attention manages or mitigates these latencies would provide better clarity for real-world implementations.

The analysis of communication volume is informative, but more practical scenarios could be discussed. For example, how does the communication overhead change when increasing not only the number of GPUs but also the number of nodes?

---

> ### Author Response · Authors · 2024-11-29
> **Answer to Reviewer yWiF**
>
> Thank you for your generally positive comments on our Tree Attention method, an efficient and exact calculation of Attention. It is true that we have developed our method by first noticing a link with energy functions and attention, which allowed us to exploit the logsumexp operation in a multi-gpu setting.
>
> You correctly summarize that Tree attention can:
> - half peak memory usage
> - achieve *8x speedups*
> - decrease communication overhead
>
> From your comments, we can also demonstrate that:
>
> -Deploying Tree Attention in a **3B-Llama** model yields a *4x speed improvement* over SOTA Ring Attention in a 3B-Llama model, thus showing that tree attention can cause *significant* speedups in decoding in full transformer models.
> -While Ring Attention is suited for TPUs, Tree Attention exploits very well other the topology of GPU systems including systems by both leading supplies – NVIDIA and AMD (RTX 4090, DGX H100 and AMD MI300X) see the Table 1 in section 6.4 and Table 2 in Appendix C3 of the revised manuscript.
> These facts mean that tree attention results in significant improvements to the efficiency of large LLM generation, especially at extended sequence lengths.
>
> We address your comments and concerns below:
>
> 1. On comparisons and real world deployment using Llama:
> For clarifications on comparisons we have added (see lines 78-81) :
>
> > Since our proposed method is an exact calculation of attention, it is a plugin replacement for any multi-GPU sequence parallel mechanism such as the state of the art Ring Attention mechanisms.
>
> Tree Attention can be immediately applied to all pretrained transformer models without other modifications. We better explain and position our method by also putting the related works from the appendix to the main paper (Section 2).
> Further, to showcase how efficiencies stack for multi-layer transformer models, we have brought results from the appendix with a 1B parameter and 3B parameter Llama model (See Table 1, also copied below). Table 2 in appendix C3 Tree Attention is almost **5x faster than Ring Attention** on RTX 4090 GPUs. Our anonymous code base has also been updated with Llama code to reproduce both ring and tree attention experiments. Thank you for suggesting this adjustment.
>
> 2. On only testing on DGX H100 systems: This is a fair point and it turns out to be a true strength of our method compared to Ring Attention. The Ring topology of Ring Attention is very well suited for TPUs but does not take advantage of the Tree topology in most other GPU systems. The table that we have copied above (see Table 1 in the paper) for Llama shows results for the NVIDIA RTX 4090, DGX H100 and the AMD MI300X. We show that Tree attention generalizes very well to other systems and in fact scales extremely well compared to the SOTA Ring Attention method on these systems.
>
> 3. Comparing against non-exact (approximate) attention methods: We would like to reiterate that our efficient Attention mechanism is an exact calculation of Attention while methods such as Linformer, Longformer or Performer are approximate methods. Since approximate methods yield further benefits at the individual device level (one GPU), we are currently investigating how Tree Attention can be combined with such orthogonal methods. This investigation will be left for future publications. We have made this point clearer in the first paragraph of the related works in Section 2.
>
> We answer your specific questions below:
>
> - Comparison to other (approximate) attention: approximate methods are generally considered orthogonal to methods that attempt to make exact attention more efficient. The combination of the two ideas is indeed very interesting and left for future work.
> - On energy functions of other (approximate) attention: As a first step, we were interested in leveraging energy functions to make the exact computation of attention more efficient. We leave the energy formulation of approximate methods for future investigation.
> - Communication volume on increasing nodes and GPUs: we have added a clarification that we are testing on an increasing number of nodes and GPUs (see lines 408-409). Since our cluster is made up of 16 nodes each with 8 GPUs, when we show in figure 3 results for 8 GPUs we are using one node, for 64 GPUs we are using 8 nodes and for 128 GPUs we are using 16 nodes.
>
>
> Thank you.

---

### Official Review · Reviewer_BUfi · 2024-11-07

**Soundness:** 3
**Presentation:** 3
**Contribution:** 3
**Rating:** 6
**Confidence:** 3

**Summary:**

The paper introduces Tree Attention, an algorithm for parallel self-attention computation across multiple GPUs to reduce communication volume and memory usage. It derives a scalar energy function for self-attention which leads to a Bayesian interpretation and a tree reduction approach for computing self-attention that is asymptotically faster due to parallelization. Experiments show close to 8× speedups on sequences of 5.12 million tokens using 128 GPUs.

**Strengths:**

1. The proposed algorithm is a novel take on self-attention and the derivation of the energy function and the associated Bayesian interpretation has the potential to contribute to a better theoretical understanding of the efficacy of self-attention in the future.

2. The tree attention algorithm cleverly exploits associativity of $\log\sum\exp$ and $\max$ to create a tree structure for attention computation in decoding that achieves logarithmic complexity as opposed to the linear complexity of prior approaches like ring attention which leads to clear latency improvements in experiments.

**Weaknesses:**

The details of the derivation of the energy function (for e.g. with the Taylor expansion in (10)) as well as the Bayesian interpretation in 3.2 seem unnecessary and may confuse the reader. If the main goal is to introduce the tree attention algorithm and its advantages, I would recommend making Section 3 more compact and moving things like the Bayesian interpretation to the discussion at the end of the paper.

**Questions:**

1. What is $d_h$ in line 101? Also there seems to be some notation abuse in the expression since it both sums over $A$ and also is apparently performed for all $A$. I think one of those should be 'a'.

2. How exactly does tree attention save inter node communications and exploit the two-level network topology? Is it by reducing the number of inter-node communications or the data volume of inter-node communications or both?

3. Why must $k_{\hat{a}'}$ , $v_{\hat{a}'} $ from neighboring devices be stored in ring attention (Page 9, lines 451-454)? Since a copy of $q$ is available at all nodes, won't it be possible to perform ring attention by just computing the flash attention computations at each node to its neighbors which can then rescale and update their attention values? This would reduce both the memory overhead as well as the communication volume ('$t$' factor in (28)) of ring attention

---

> ### Author Response · Authors · 2024-11-29
> **Answer to Reviewer BUfi**
>
> Thank you for the generally positive review of our Tree Attention paper. We agree that the link between Energy functions, its associated Bayesian interpretation, and attention is an interesting one. The associativity of $\log⁡∑\exp$ derived from the link to Energy functions allowed us to build the Tree attention algorithm and unlock efficiencies in speed, memory and communication volumes during multi-GPU LLM generation.
>
> You have left valuable feedback that we have incorporated in the current version of the manuscript. We hope that following changes address all your concerns:
> 1. We have moved many of the details of the energy function derivations and Bayesian interpretation to *Appendix C* to streamline the main flow of argument and focus more on the empirical results in the main paper body. Our Sections 3 and 4 are now more compact.
> 2. We have added more details in the new version of the manuscript (which we also answer below) to expand on (a) how tree attention saves on inter-node communication, (b) why must $ka^′$, $va^′$ come from neighbouring devices for ring attention.
>
>
>
> We rewrite directly our clarifications following your questions below:
> 1. on A and dh in line 101: thank you for pointing this typo out. We have removed the for all $A$ in {1 to $d_h$} and moved the dot product equation to the appendix since it is a commonly known equation.
> 2. On Tree Attention inter-node communication: Tree attention saves on both inter-node communication volume and number of communications. Tree attention exploits a bandwidth/latency tradeoff to make overall attention communication more efficient. The communication volume is the numel; for ring attention numel is $2btd$ in equation 27 which is higher than the numel of $bd + 2bn_h$ for tree attention in equation 29. At the same time, the number of communications is p for ring attention which is higher than $2(p-1)/p$ for tree attention. We have clarified this point in the manuscript (see lines 453-475).
> 3. Ring attention $ka^′$, $va^′$ from neighbouring devices: The single query has to see every key and value - this can be accomplished either (1) by moving the key and value shards, which is what ring attention does, or (2) by communicating the partially computed result of the attention operation - which is what tree attention does. The strategy of communicating neighbouring partial results (as you had mentioned) is indeed our solution with Tree Attention. We are happy to discuss this further should you have any other questions.

---

> > ### Comment · Reviewer_BUfi · 2024-12-03
> > **Re**
> >
> > Thank you for your response. As I had already recommended accepting the paper, I will keep my score unchanged. I do not have any other questions or concerns.

---

### Official Review · Reviewer_aspw · 2024-11-08

**Soundness:** 2
**Presentation:** 3
**Contribution:** 2
**Rating:** 6
**Confidence:** 3

**Summary:**

The manuscript "Tree Attention: Topology-Aware Decoding for Long-Context Attention" proposes a novel strategy for computing self-attention coefficients in Transformers, the fundamental operation behind the functioning of this highly celebrated generative model. The method is based on two fundamental ideas. The first is theoretical, inspired by statistical mechanics: at each step of the data generation process, self-attention can be computed as the gradient of a function that resembles the free energy of a multi-particle system at thermodynamic equilibrium. The second idea, on the other hand, concerns hardware: an ensemble of GPUs can be employed to compute the energy function along a tree-like scheme. Consequently, self-attention is computed in such a way that inter-node communication is avoided, unlike in recent architectures presented in the literature.
The main result of the paper is an algorithm for computing self-attention that is faster, as it scales logarithmically with the number of GPU nodes used in the process; requires less memory for storing useful variables before usage; and reduces inter-node communication volume for shorter execution times.

**Strengths:**

The main strength of this work lies in the elegant simplicity of the energy-driven theoretical argument, which enables a simplification of the architecture used to compute self-attention coefficients. The paper is well-written and pedagogical, even for readers who are not experts in physics or in the technical aspects of executing self-attention computation on this type of machine.

**Weaknesses:**

I believe that the authors should make a stronger effort in validating the theoretical predictions about the improvement in the computational cost of self-attention proposed by the Tree Attention method, which is also their main scientific question. The scaling of the execution time $T$ with respect to variables $N$ and $p$ is reported in figure (3), without any proper experimental analysis of the results, which would effectively validate the theoretical predictions and convince possible future users. While their report only shows that Tree Attention is faster than Ring Attention, a deeper analysis of the experiments would be definitely needed: e.g. one might start from a linear fit of $T$ Vs. $N$ in linear scale and a linear fit of $T$ Vs. $p$ in log scale; a proper expected scaling (to be confirmed by the plots) of Ring Attention is never provided, while the condition $T_{ring} > T_{tree}$ might give useful bounds on the choice of the parameters of the system.
Moreover, while the paper emphasizes a strong connection with modern Hopfield models (which are even mentioned in the abstract), this aspect is never properly addressed in the manuscript. As a matter of fact, the only—yet fundamental, for the sake of the results—physics-inspired part of the work consists in deriving self-attention by applying the logsumexp function, which satisfies the associative property. On the other hand, self-attention has emerged in previous papers (see: Hoover et al. 2023, Kozachkov et al. 2023, D’Amico et al. 2024) due to its analogy with the gradient descent dynamics over the energy function of the modern Hopfield model. Since both the dynamic aspect/analogy part and the thermodynamic one, which involves considerations about minimization of a free-energy function, are only briefly discussed, I would consider sacrificing some of the discussion about Hopfield models to give more room to a broader test of the analytical predictions.

**Questions:**

I will now present some comments and questions to the authors, aimed exclusively at improving their interesting work about a more effective computation of self-attention in transformers. Comments and questions are listed here below, divided in Major issues, Minor issues and Typos.

Major issues:

1) From Theorem (1), the computational complexity of self-attention, for a single query, is $O( \frac{N}{p} + \log p )$  where $N$ is the full length of the sequence and $p$ is the number of processors. The total complexity is thus quadratic in $N$. There are examples of methods that simplify the total complexity to $O(N)$ (e.g. BigBird, Longformer) or $O(N \log N)$ (e.g. Reformer) which, by the way, do not rely on a modification of the processors disposition and are not necessarily exact. If we assume the number of GPUs to scale as $p = \alpha N$ where $\alpha > 0$, then the complexity per query would scale as $O(\log N )$, hence the total complexity as $O(N\log N)$ which equals the one obtained by Reformer. Nevertheless, even if this scenario appears difficult to reproduce, since $N$ is large by definition, this limit is mentioned in the text (row 269). Could the authors comment on this?

2) Regarding the experiments reported in Section 5.1. Physical limitations seem to impose that $p = O(1)$, and $p \ll N$. As a consequence, the effective behaviour of the execution time $T$ with respect to $N$ is $T \simeq \frac{N}{p}$, hence it is linear in $N$. On the other hand, when $N$ is fixed, we have $T \simeq  \frac{N}{p} + log(p)$, it should then scale logarithmically with $p$ for large $p$. Can the authors show these trends more clearly from figure (3) e.g. increasing the number of points (in the intermediate range, i.e. between the current minimum and maximum values on the x-axis) and performing a proper fit of the data-points? Moreover: what is the expected scaling of the computational cost of Ring Attention as a function of $(N,p)$ ? It seems from figure (3,b) that the behaviour of $T_{ring}$ as a function of $p$ is linear, while in principle it should be constant and remain $T_{ring} \gg \log p$. If not, please provide for a theoretical argument to explain this behaviour.

3)May the authors show that the formulation of $o$ in line (2) of Algorithm (3) already normalized by $lse$ is equivalent to implementing the formulation from line (4) in Algo (2)? Expanding the mathematical passages in Algo (3) might useful to visualize this aspect.

4) The difference between the memory costs of Ring and Tree attentions is $\delta_{M} = Mem_{ring} - Mem_{tree} = 2b(\frac{N}{p}\cdot d - n_{h})$. Could the authors check for the scaling of this quantity with respect to $(N,p)$ to satisfy $\delta_{M} \propto \frac{N}{p}$ ? I believe this would make the analysis more rigorous and reliable.

5) I propose to the authors to add the most recent attempt to map the self-attention mechanism into a energy-based model, i.e. D’Amico et al. 2024, to the Related Work appendix. Nevertheless, i would spend much more space in the same paragraph to mention previous works in the matter of the computational complexity of computing self-attention, even the cases where multiple processors were not employed. A useful review about this problem can be found in Keles et al. 2023.

Minor Issues:

6) I suggest the authors to substitute the Wikipedia reference for the definition of Helmholtz free-energy with a more reliable and solid scientific source, among the thousand great statistical mechanics books. One example is  Landau, Lifshitz, "Statistical Physics".

7) The index A in the second formula in section 2 (i suggest to add numbers to all formulas) seem to be redundant, including the subsequent comment "The capital Latin indices (𝐴) span the hidden dimension $𝑑_ℎ$". I would simply substitute the letter "d" on top of the sum with $d_h$.

8) What is the difference between "Absolute" and "Relative" execution time in Section 5.1?

9) Can the authors add a definition for the batches $b$ in Section 5.2 ?

Typos:

10) Row 144: Cumulant.
11) Row 505: Its communication.

**Details Of Ethics Concerns:**

No concerns related to ethics found in the paper.

---

> ### Author Response · Authors · 2024-11-29
>
> Thank you for taking the time to read our paper and provide valuable feedback. We have taken your recommendations and made the following changes to the manuscripts:
> 1. We have moved the link to energy functions to the appendix to leave more room for empirical results and discussions (see Sections 5.3 and 6.4).
> 2. We have toned down the theoretical links to Hopfield Networks in abstract and introduction.
> 3. We have added a related works section that better positions our paper as method to parallelize exact attention across multi-GPUs
>
>
> We also fixed the typos, minor clarification and added references that you have highlighted, especially those relating to other energy based model interpretations of self-attention. We really appreciate your comments on this.
>
> Please find below direct responses that should address your specific questions and comments:
>
> 1. On comparing the Reformer $O(N\log⁡N)$ complexity with Tree Attention: As you correctly noted, each Tree Attention query has time complexity of $O(\log⁡N)$ per query, assuming $N$ processors. This is indeed comparable to the time complexity of approximate attention methods such as the Reformer. However, the Reformer achieves this reduced time complexity by modifying where the attention is done using locality-sensitive-hashing thus changing the attention calculation. Tree attention does not change the attention calculation and gives numerically *identical* outputs to standard attention. Tree attention is a new way to parallelize the exact attention calculation. As such tree attention can be applied exactly to existing pretrained models while the Reformer either requires retraining from scratch or a finetuning phase. As such, the two methods are *orthogonal*. Nonetheless, we have plans to apply our method to approximate attention methods in future work since we see clear benefits in using both techniques together.
>
> 2. Linear fits of $T_{ring}$ and $T_{tree}$ vs. $N$:
> The main caveat to matching the theoretical time complexity formulae to the experiments we ran is that the former does not account for time spent communicating partial results, which in practice dominates the actual execution time. The reason ring attention sees a linearly rising execution time as a function of the number of devices is because the time spent communicating the shards of $K$ and $V$ grows with the number of devices between which we communicate them. We appreciate the point raised, and will emphasise this caveat in the revision of the paper (see Section 5.3).
>
> 3. We have clarified in the manuscript how $O$ (now just called lse everywhere) is used to normalize $R_a$ in Algorithm 2, similarly to how lse is used for normalization in Algorithm 3. Please see line 2 and line 5 in Algorithm 2 (which points to $lse = Reduce(logsumexp, 𝑟)$ in Algorithm 1). We have also changed the titles of the algorithms boxes to make it clearer that the only difference between Algorithm 2 and Algorithm 3 is that algorithm 3 uses Flash Attention. Thank you for pointing this out.
>
> 4. The memory trends are plotted in Figure 4. We see from the plots that as “predicted by theory, scaling hidden size or sequence length scales Ring Attention peak memory usage about *2x* faster than Tree Attention” (see lines 447-449). We observe in equations 8 and 9 that memory does increase *2x* faster as well. We are happy to discuss this point further.
>
> 5. It is true that a related works section would serve to better position our paper. We have added related works in section 2 that discusses past work to improve attention execution time and memory via (a) approximate attention methods, (b) parallelization on a single device and (c) multi-device parallelization. We have also added mentioned citations. Thank you again for your feedback.
>
> 6. Figure 3: Relative execution time is the execution time indexed at the ring attention execution time when the sequence length is 80k tokens. While this is described in the legend of the figure 3, we have added the following (see lines 374-405):
>
> > To better highlight execution time trends with an increasing sequence length, we have also added relative execution time of both methods with respect to the execution time of ring attention at a sequence length of 80k. With relative execution time in Figure 3a, we notice that Tree attention's execution time flattens as the number of GPUs increases, while Ring Attention relative execution time continues to increase.
>
> 7. I would simply substitute the letter "$d$" on top of the sum with $dh$: You are absolutely correct and thank you for pointing this out. We have applied your recommended fix and moved the definition of the dot product in attention to the appendix since it is already a well known formulation.

---

> > ### Comment · Reviewer_aspw · 2024-11-29
> >
> > I thank the authors for addressing my comments and questions about their work. I can acknowledge a net improvement in the presentation of the results and a deeper focus on their main research question, i.e. the complexity of attention computing.
> > I also recognize that Tree Attention computes the attention coefficients exactly, while several methods in the literature imply a modification in the computation that works as an approximation.
> > Nevertheless I'm still not very much satisfied by their answer to point 1, specifically by the statement
> >
> > > each Tree Attention query has time complexity of O(log⁡N) per query, assuming N processors.
> >
> > I do not see how this scenario of $p=O(N)$ might be implemented in practice. In fact, the experiments provided in Section 6.1 all rely on $p \ll N$. If the authors are able to perform an experiment and show that the total complexity scales as $\mathcal{O}(N log(N))$ they should give evidence of it in the paper, otherwise I would conclude that Tree Attention cannot be considered faster than $O(N^2)$ and the only gain from the tree structure of GPUs is a factor $1/p$.
> > For this reason, I thank the authors again for the discussion but I will keep my rating as it is for now.

---

> > > ### Author Response · Authors · 2024-12-02
> > > **Thank you Reviewer aspw for your follow-up. We added clarifications and made clear our assumption in Theorem 1.**
> > >
> > > Thank you for your follow up on this. We now better understand the two main issues at hand and we will answer them individually below:
> > > 1. **Tree Attention Complexity:** You are correct to say that the *"gain from the tree structure of GPUs is a factor $1/p$"* and that for a query of size $N$ we would indeed have an $O(N^2 / p)$ complexity.  For a query size $= 1$ in the cases that we study, Vanilla Attention complexity is $O(N)$. With Tree Attention parallelization, $O(N/p)$ leads the complexity equation when $p << N$.
> > >
> > > We will also add to the manuscript:
> > > > For $p << N$ and in the absence of communication costs, Ring attention is also $O(N/p)$. In practice however, we see that Tree Attention is able to better reduce communication costs by exploiting the hierarchical network structure and calling into a tree reduction operation between nodes. This is because the tree algorithm has a lower communication volume than ring attention applied to decoding (see Section 5.2).
> > >
> > > 2. **Getting $O(N \log N)$ for $p = O(N)$:**  This is indeed a case that is unlikely to happen in practice, especially for a large sequence length N and when there is a cost of communication. While, highly parallelized max and  $\log\sum\exp$ have been studied in the past [1, 2] where the $O(N \log N)$ complexity has been proven with toy examples on CPUs, our setup is a different.  As we mention in *Point 2* of our rebuttal, time complexity estimates do not perfectly map to execution time because the latter is bottlenecked by cost of large tensor communication in a multi-GPU setting. While we discuss this point in the paragraph titled *Communication overheads* (see lines 286-316), we will make it clearer the underlying assumption in *Theorem 1*:
> > >
> > > > Theorem 1: **Assuming instantaneous data movement**, the time complexity of a reduction operation involving an associative function, such as $logsumexp_𝑎$ or $\max_𝑎$, [...]".
> > >
> > > We will also add the manuscript on lines 282-283:
> > > > While our theoretical analysis points to an overall complexity of $𝑂(3(𝑁/𝑝 + log 𝑝))$,  it is not achievable in practice, especially when data movement is accounted for."
> > >
> > > *References*:
> > >
> > >
> > > [1] Single-pass Parallel Prefix Scan with Decoupled Lookback, Duane Merrill and Michael Garland, 2016
> > >
> > >
> > > [2] Prefix Sums and Their Applications, G. Blelloch. in Sythesis of parallel algorithms, Morgan Kaufmann Publishers Inc., (1990)
> > >
> > > Should you have any other comments on the added assumptions and clarifications, we are happy to discuss further.
> > > Thank you.

---

> > > > ### Comment · Reviewer_aspw · 2024-12-02
> > > >
> > > > I thank the authors for their revision and clarifications, that I will keep in mind for further discussions.

---

### Author Response · Authors · 2024-11-29
**General Rebuttal**

We would like to thank all reviewers for your constructive feedback. Compared to Ring Attention, a *SOTA method* to parallelize LLM exact attention across multiple GPU, we agree with most reviews that the key strengths of Tree Attention are:
1. lower memory overhead by a factor of 2
2. faster throughput (8x faster)
3. less communication volume (2x less)
4. better scaling with increasing number of GPUs (logarithmic scale)

We have used your comments to add clarifications and streamline the presentation of the material. Here is a list of the main changes made to the paper. We have:
1. Added Table 1 in Section 6.4 showing that Tree Attention can be used in a Llama 3 model and achieves **x3-4 speedups** compared to Ring Attention.
2. Added Related Works in section 2 to better position Tree Attention as multi-GPU parallelization of the exact computation of Attention that belongs to the *same class of algorithms as Ring Attention*.
3. Compressed our derivation of Attention as the gradient of an energy function in Section 4 and moved the theoretical links to Hopfield Networks and the Bayesian interpretation to Appendix C.
4. Made it much easier to follow how the energy formulation using the $\log\sum\exp$ allows us to find an efficient algorithm in Section 5.

---

### Meta-Review · Area_Chair_Z71Z · 2024-12-12

**Metareview:**

The paper proposes a new method (Tree Attention) for parallel self-attention computation across multiple GPUs. This leads to a reduction in communication overhead and memory usage, as compared to existing approaches such as Ring Attention. The idea is to derive a scalar energy function for self-attention, which is computed efficiently due to parallelization.

The reviewers appreciated the elegant simplicity of the theoretical argument, the reduction in memory usage and the parallelism of the proposed approach. However, during the review process a number of issues were raised concerning the performance of the method, its evaluation and using ring attention as a baseline. These concerns were not addressed in a satisfactory way during the rebuttal process. In the final discussion between AC and reviewers, reviewers Eqp7, aspw and mxBy agreed that the paper, in its current form, does not provide sufficient evidence of a significant improvement over the state of the art. I agree with this assessment and therefore recommend rejection at this stage.

The reviewers did praise the novelty of the approach, which I also find valuable. I would thus encourage the authors to resubmit an improved version that addresses the concerns raised during this review process to a future venue.

**Additional Comments On Reviewer Discussion:**

Reviewer Eqp7 summarized three important and unresolved weaknesses: Ring Attention as a baseline; concerns regarding evaluation and implementation; unsatisfactory performance. Reviewers aspw and mxBy also agreed that the paper does not provide sufficient evidence of its superiority w.r.t. existing work. I agree with these concerns, which leads me to a 'reject' decision.

---

### Decision · Program_Chairs · 2025-01-22

Reject